# Wastewater Analyses for Psychoactive Substances at Music Festivals: A Systematic Review

**DOI:** 10.3390/bs15121672

**Published:** 2025-12-03

**Authors:** Ringala Cainamisir, Xiao Zeng, Samuel B. Himmerich, Hubertus Himmerich

**Affiliations:** 1Department of Psychological Medicine, Institute of Psychiatry, Psychology, and Neuroscience, King’s College London, London SE5 8AF, UK; ringala.cainamisir@kcl.ac.uk (R.C.); xiao.1.zeng@kcl.ac.uk (X.Z.); shimmeri@on.wwu.de (S.B.H.); 2School of Business & Economics, University of Münster, 48149 Münster, Germany; 3South London and Maudsley NHS Foundation Trust (SLaM), London SE5 8AF, UK; 4Bundeswehr Center for Military Mental Health, Military Hospital Berlin, 10115 Berlin, Germany

**Keywords:** recreational drugs, illicit drugs, new psychoactive substances, psychoactive pharmaceuticals, music festival, wastewater, pooled urine, wastewater

## Abstract

Music festivals have emerged as venues for the consumption of recreational drugs and novel psychoactive substances. This systematic review provides the first critical evaluation and synthesis of published wastewater analyses for detecting recreational drug use at music festivals worldwide. Following Preferred Reporting Items for Systematic Reviews and Meta-analyses (PRISMA) guidelines, we systematically searched PubMed, Embase, MEDLINE, and SCOPUS databases using terms combining music festivals, drugs, and wastewater analysis. Twenty-three studies were included, spanning festivals with 6200 to 600,000 attendees. Two primary sampling approaches emerged: wastewater sampling (16 studies) and pooled urine sampling (7 studies), using Liquid Chromatography–Tandem Mass Spectrometry or Liquid Chromatography–High-Resolution Mass Spectrometry for chemical analysis. 3,4-Methylenedioxymethamphetamine (MDMA; ecstasy) emerged as the most consistently detected substance. Regional variations included a dominance of methamphetamine in Eastern Europe, MDMA use in Western Europe, and a high prevalence of cocaine use in South America. Regarding the music genre, electronic dance music events showed markedly higher MDMA rates. Limitations include geographic underrepresentation of African and Asian countries and a gender bias in pooled urine sampling. Future research should work on enhancing sampling infrastructure, analytical capabilities, public health surveillance, and harm reduction strategies.

## 1. Introduction

### 1.1. Psychoactive and Illicit Drugs

A drug is defined by the American Psychological Association (APA) Dictionary of Psychology as any non-food substance that affects physical or mental processes, often used for medical but also recreational purposes ([3]). Pharmaceutical manufacturing is regulated by agencies like the [62] ([62]) and the [19] ([19]) to ensure compliance with good manufacturing practices. While most drugs are used as intended, people use them non-medically for their psychoactive effects ([55]). Many pharmaceuticals are naturally derived from plants, while others are entirely synthetic ([43]; [34]).

Drugs that are manufactured for medical purposes yet not used for their approved indication, alongside substances created solely for recreational purposes, are often referred to as “illicit drugs”, or “drugs of abuse” ([28]). However, these terms focus on the current legal status of the drug, but not its effects or other relevant aspects. Therefore, we will refer to non-prescribed drugs as “recreational drugs.” The primary categories encompass hallucinogens, cannabinoids, synthetic stimulants, and tranquillisers ([18]; [42]). In epidemiological research, recreational substance use is often examined through surveys ([57]; [63]). Self-reports exhibit notable limitations, including the potential for underreporting or over-reporting of actual drug use. Aspects such as restricted recall and socially acceptable responses may lead to incorrect reporting ([32]; [31]). Selection bias can arise from non-random recruitment of participants or from certain subgroups having a reduced likelihood of participation ([25]; [33]), and users may be unaware of the specific substances they have ingested ([17]; [59]). The emergence of numerous New Psychoactive Substances (NPS) on the drug market may have exacerbated this issue in recent years ([22]). Drug abuse, whether involving traditional pharmaceuticals or NPS, can progress to substance use disorder, a clinical condition characterised by significant impairment and distress. According to the Diagnostic and Statistical Manual of Mental Disorders, Fifth Edition (DSM-5), substance use disorder is diagnosed when an individual meets at least two of eleven criteria within a 12-month period, including impaired control over substance use, social impairment, risky use, tolerance development and withdrawal symptoms ([2]).

### 1.2. Wastewater Analysis for Psychoactive Substances

The presence of recreational drugs in the environment is gaining acknowledgement as a significant concern. The initial implementation of wastewater analysis aimed to scrutinise the environmental pollution resulting from these substances and to evaluate the efficacy of facilities responsible for dealing with their elimination ([60]). Recent investigations have demonstrated how a similar methodology may be applied to quantify recreational substances found in wastewater. Wastewater-based epidemiology (WBE) is a population-level monitoring approach that utilises the examination of chemical and biological indicators in wastewater to assess community-wide exposure to diverse substances, including recreational drugs, pharmaceuticals, and pathogens. It enables non-invasive surveillance of substance use trends and public health indicators in defined populations ([24]; [46]); therefore, it is a method commonly utilised by various researchers and organisations as an auxiliary measure of drug consumption, including the European Monitoring Centre for Drugs and Drug Addiction (EMCDDA) ([17]; [46]; [24]; [45]). Indeed, the quantities of recreational substances utilised globally are on par with those of medicinal drugs ([69]). The inspection of biological specimens from individuals is costly, labour-intensive, and necessitates consent ([9]). Samples from pooled urine obtained from portaloos and movable toilets provide benefits over wastewater by facilitating sample collection closer to the site of elimination ([4]). This proximity minimises uncertainties such as transport within the sewer system and dilution from runoff water during wet weather ([9]).

### 1.3. Psychoactive Substances at Music Festivals

Current literature suggests that music festivals can be venues where novel NPS and other recreational drugs are consumed ([14]). Research indicates that attendees of music events engage in the consumption of recreational substances at higher rates compared to similar age groups in the general population ([27]; [37]; [66]). Such events provide an advantageous setting for monitoring recreational drug use among populations at increased risk of substance-related harm ([67]; [64]; [36]).

One example of a major music festival is the Glastonbury Festival, which is held in Somerset, England, and is deemed the world’s largest greenfield music and performing arts festival. In 2019, when the Glastonbury Festival had 203,000 attendees, researchers tested for recreational drugs, including cocaine and MDMA, in the Whitelake River near the festival site and in the neighbouring Redlake River. Both rivers were sampled before, during, and after the event. Cocaine, its metabolite benzoylecgonine (BZE), and MDMA were detected at all sites, but concentrations and mass loads were significantly higher downstream of the festival site. MDMA mass loads were 104 times greater downstream, and cocaine/BZE loads were 40 times higher. The highest MDMA concentration (322 ng/L) was recorded the weekend after the festival and was deemed harmful to aquatic life ([1]).

### 1.4. The Potential of Wastewater Analyses for Psychoactive Substances at Music Festivals

WBE can make use of mass gatherings, such as music festivals, as these events are focal points for recreational drug consumption, encompassing a diverse assortment of substances such as MDMA, cannabis, cocaine, ketamine, and a growing variety of NPS ([37]; [26]; [22]).

Moreover, such analyses can provide emergency responders, A&E clinicians, and psychiatrists with insights into the chemicals likely to be implicated in acute medical occurrences during musical events, thus enhancing clinical readiness and response ([4]; [17]). For instance, research examining wastewater from six European festivals identified several narcotics and NPSs, such as synthetic cathinones and phenethylamines ([9]). This information can inform medical personnel of the drugs currently used by some attendees of music festivals. Additional examples may arise from the psychiatric domain, where several chemicals can elicit acute psychiatric symptoms, including hallucinations or psychosis ([20]; [54]; [50]).

WBE might detect specific drug use patterns associated with the musical genre, the cultural milieu and the geographic region ([46]; [68]; [26]; [9]). However, regarding the causality, the association between the consumption of a specific drug with a musical genre might be an epiphenomenon that is driven by the social context, not the type of music per se.

Wastewater analysis can contribute valuable data for public health surveillance, and this information might hopefully support the development of more effective, evidence-based policies and improve preparations for medical and psychological support for musical events with harm reduction as the central focus.

### 1.5. Aim

Although wastewater-based epidemiology at special events has been previously examined, existing reviews have adopted a broad approach encompassing multiple types of events. For instance, [65] ([65]) included music festivals alongside sporting events, holidays, and site-specific studies (e.g., educational institutions and prisons) in their systematic review.

A comprehensive systematic review of scientific studies on wastewater analyses for psychoactive substances specifically at music festivals has never been conducted. Therefore, the aim of this systematic review is to critically evaluate and synthesise the existing literature on wastewater analysis to measure recreational drug consumption at music festivals, providing an in-depth examination in this specific context of festivals where music is the main feature.

## 2. Materials and Methods

This systematic review was conducted according to the PRISMA guidelines ([49]), and the PRISMA elements were documented according to the [51] ([51]) checklist (PRISMA 2020 Checklist, see Appendix A: PRISMA 2020 Checklist). To provide a systematic and repeatable process, each stage in this evaluation was predefined and documented. The systematic review was registered with the Open Science Framework ([47] ([47]); https://osf.io/9aq7p/ (accessed on 22 August 2025)).

### 2.1. Search Strategy

As the topic of this systematic review has a focus in biomedical, pharmacological and toxicological research, we chose the databases [52] ([52]), [16] ([16]), MEDLINE (via [48] ([48])), and SCOPUS as data sources for this review. PubMed is a collection of articles covering life sciences, biomedical science and medicine; Embase covers biomedical research with an emphasis on drug research and pharmacology; MEDLINE is a comprehensive database of references with a strong focus on evidence-based clinical and biomedical research; and SCOPUS is a multidisciplinary database with comprehensive global coverage of peer-reviewed literature across the sciences, technology, medicine, social sciences, and humanities.

These electronic databases were used in the literature search to create a list of pertinent studies. Keywords that operationalise the research question were used to generate search terms. The search was enhanced by using Boolean search operators like “AND” and “OR”. To find all publications that are appropriate for responding to the research topic, the following search phrases were used: (“music” OR “music festival” OR “concert” OR “music performance”) AND (“drugs” OR “drug misuse” OR “illicit drugs” OR “psychoactive pharmaceuticals” OR “new psychoactive substances” OR “ayahuasca” OR “cannabis” OR “marijuana” OR “pot” OR “weed” OR “benzodiazepines” OR “cocaine” OR “fentanyl” OR “GHB” OR “psychedelics” OR “cocaine” OR “hallucinogens” OR “heroin” OR “inhalants” OR “ketamine” OR “khat” OR “kratom” OR “LSD” OR “MDMA” OR “Acid” OR “Ecstasy” OR “Molly”) AND (“wastewater” OR “Wastewater treatment plants” OR “sewage” OR “effluent” OR “run-off water” OR “foul water” OR “toilets” OR “portable toilets” OR “porter potty” OR “portaloo” OR “pooled urine”). Following the initial search, Endnote and Ryyan were used to identify and eliminate duplicate studies.

### 2.2. Inclusion and Exclusion Criteria

Finding and reviewing research data on wastewater analyses for psychoactive and recreational substances at music festivals was the goal of this search. The following predetermined eligibility criteria were developed:

Inclusion Criteria:Research in humans;Either a meta-analysis or an original article (generating new research data);Research on wastewater analyses for drugs during/after music festivals, which means that music was the main feature of the festival;Published in English, German, or Romanian.

Exclusion Criteria:Case reports and case studies;Animal and cell studies;Review articles;Perspective papers;Letters (without data);Master or doctoral theses.

### 2.3. Screening/Selection Process

Following the completion of the literature search and the removal of duplicates, a preliminary evaluation of the abstract and title was conducted for the studies, followed by a thorough examination of the entire content as part of a two-stage screening procedure. To help ensure strong inter-rater reliability, two independent review authors (RC and XZ) co-screened the studies to determine if the eligibility criteria were met. Every step of the authors’ decision-making process was documented using Word Documents. Articles that were excluded were listed separately, along with any justifications for their exclusion. A third independent reviewer (HH) was consulted, if a consensus could not be established after the two reviewers had discussed and tried to resolve any disagreements. Furthermore, in accordance with the [29] ([29]) for Analytical Cross-Sectional Studies, the two reviewers who extracted the data (RC and XZ) conducted analyses of study quality and risk of bias independently (see Appendix A; questions 3 to 7 of the JBI Critical Appraisal Tool cover information pertaining to the risk of bias). Any disagreements were reconciled with HH, the third rater.

### 2.4. Data Extraction

Key information from each manuscript was gathered through data extraction. The predetermined categories consisted of (1) citation characteristics (e.g., authors, year of publication, country of publication); (2) sample characteristics (festival goers, music genre, number of participants, open air or in-house festival, age and gender if reported); (3) methodological characteristics (wastewater analysis methods, collection of wastewater, analytical techniques); (4) outcome characteristics (drugs and drug concentrations).

### 2.5. Ethical Considerations

Systematic reviews are not typically required to obtain ethical approval, as they do not produce original literature or directly handle participant confidentiality, in contrast to primary research. Furthermore, only studies that had already been published and received the necessary ethical approvals were used in systematic reviews. However, in the event that any ethical issues arose, we would have adhered to the procedure specified in the British Psychological Society (BPS) Code of Human Research Ethics ([44]) and the Medical Research Council’s (MRC) Principles and Guidelines for Good Research Practice ([61]).

### 2.6. Analysis

We summarised the included studies in Table 1 and analysed their content narratively according to their citation, sample, methodological and outcome characteristics.

## 3. Results

### 3.1. Search Results

A total of 200 records were discovered using the predefined search strategy across four databases: PubMed (n = 33), Embase (n = 39), MEDLINE (n = 10), and SCOPUS (n = 118). Following the removal of 20 duplicates in EndNote and an additional 58 duplicates in Rayyan, 122 entries were retained for title and abstract screening. Out of these, 33 articles were evaluated at the full-text level, and out of 9 excluded, 6 were discarded due to the unavailability of the full text despite our attempt to contact the corresponding authors. Ultimately, 23 papers met the inclusion and quality criteria and were incorporated in the final review. Figure 1 displays the PRISMA Flow Diagram, which delineates the screening and selection procedure. The research articles presented are elaborated in Table 1, which outlines the study characteristics, country-level drug trends, and the analytical techniques utilised.

### 3.2. Study Characteristics

This review included data from 23 studies published between 2010 and 2024. The studies were conducted across multiple countries, with the highest number coming from Australia (n = 5) and Slovakia (n = 4), followed by Belgium (n = 3). Two studies each were comprised from Croatia, Norway, Spain, Switzerland, and the United Kingdom, while single studies were also included from Brazil, Czech Republic, Denmark, France, Hungary, Portugal, Serbia, and Taiwan.

### 3.3. Study Results

#### 3.3.1. Sampling Methods

The choice of a suitable sampling approach is a critical component of WBE, especially during high-intensity social events such as music festivals. Two primary sampling methods were utilised: wastewater samples collected from sewage systems and pooled urine samples obtained directly from portable facilities.

Wastewater sampling approaches were employed in 16 out of 23 studies, representing the most common sampling strategy for festival-based substance monitoring. The predominant method was 24 h composite sampling, utilised in 14 studies, which involved automated collection systems that gathered samples at regular intervals throughout the festival period to account for temporal variations ([9], [10]; [8]; [39], [40]; [56]; [5]; [7]; [12]; [11]; [23]; [38]; [41]; [53]; [58]). Time-proportional sampling was the preferred composite approach, typically collecting smaller volumes at 15 min intervals over 24 h periods ([12]; [40]; [41]; [56]).

Automated sampling devices were extensively used to ensure consistent sample collection under controlled conditions. These systems typically operated in refrigerated environments to preserve sample integrity and prevent degradation of target analytes ([5]; [8]; [10]; [9]; [12]; [15]; [23]; [36]; [40]; [39]; [53]). The sampling frequency varied between studies. Several studies took samples every 15 min. [53] ([53]) used a battery-powered sampling device which collected ~0.1 mL subsamples every 10 s into one composite sample over the 24 h.

Grab sampling of wastewater was employed in a limited number of studies, particularly where composite sampling was not feasible due to logistical constraints or when investigating specific temporal patterns ([30]).

Pooled urine sampling was utilised in 7 studies, offering advantages in terms of higher analyte concentrations and reduced sample dilution compared to wastewater matrices ([9]; [13]; [21]; [22]; [26]; [35]; [58]). Samples were collected from portable toilets, urinals, and storage tanks using grab sampling techniques. The sampling process typically involved drawing samples from internal storage tanks connected to male urinals or portable toilets using 50 mL syringes ([9]; [35]; [58]), 50 mL pipettes ([13]), or pumps ([21]; [26]).

Some studies employed hybrid approaches combining multiple sampling strategies, such as the simultaneous collection of wastewater and pooled urine samples, to maximise detection capabilities and provide complementary data sets ([9]). [8] ([8]) extended wastewater monitoring to include supplementary surface water (river and lake water) to assess broader environmental contamination patterns beyond the immediate wastewater treatment infrastructure, demonstrating the downstream fate of drugs detected in festival wastewater across multiple Swiss cities. Additionally, one study incorporated self-reported surveys and oral fluid samples into the data collection to reveal discrepancies in reported vs. actual drug use ([22]).

#### 3.3.2. Analytical Methods

A variety of analytical methods were employed across the included studies to identify and quantify psychoactive substances in wastewater and pooled urine samples (see Table 2). The choice of analytical techniques was primarily driven by three factors: target analyte classes (traditional drugs versus NPS), sample matrix characteristics (wastewater versus pooled urine), and available laboratory infrastructure.

LC-MS/MS emerged as the predominant quantitative approach (16/23 studies), chosen for its established reliability in detecting traditional recreational drugs at trace levels. This preference reflected researchers’ confidence in validated methods, with studies that focused on consumption estimation consistently selecting this approach for its quantitative precision. The approach is also the most selective for targeted analysis.

LC-HRMS adoption in five studies specifically addressed NPS detection challenges, recognising that the rapid emergence of novel compounds made targeted approaches insufficient. Researchers chose high-resolution techniques when study objectives included comprehensive screening or when preliminary intelligence suggested significant NPS activity.

Solid-phase extraction (SPE) was employed in 17/23 studies, becoming standard for wastewater analysis to address the highly diluted nature of analytes. However, both [41] ([41]) and [8] ([8]) demonstrated that direct injection methods could achieve comparable sensitivity for smaller populations and for multi-site wastewater monitoring, respectively, challenging the assumption that SPE was universally necessary. However, this technology is not widely available to other labs. Critical comparative findings shaped subsequent methodological choices. [26] ([26]) demonstrated immunoassay inadequacy for pooled urine screening, detecting 77 drugs and metabolites via Ultra-Performance Liquid Chromatography High-Resolution Time-Of-Flight Mass Spectrometry (UPLC-HR-TOF-MS) compared to limited immunoassay detection due to cut-off value differences and cross-reactivity issues.

Evidence-based optimal combinations emerged from comparative studies: SPE + LC-MS/MS provided the gold standard for quantitative analysis of traditional drugs, while SPE + LC-HRMS offered superior capabilities for NPS detection and confirmation. Pooled urine combined with LC-HRMS delivered the highest sensitivity for real-time festival monitoring, though with geographic and demographic limitations. Cost-effectiveness considerations influenced method selection, particularly for studies requiring extensive temporal sampling or multi-site coordination ([35]; [9]). However, the direct injection approach demonstrated by [8] ([8]) offered advantages for multi-city monitoring programmes.

#### 3.3.3. Drug Preferences and Patterns

Wastewater analyses revealed distinct patterns in psychoactive substance consumption at music festivals, with consistent preferences emerging for specific drug categories despite variations in geographical locations, festival types, and analytical methodologies.

MDMA emerged as the most prominently detected substance across multiple festival settings and geographical regions. [36] ([36]) documented MDMA loads averaging 240 mg/day/1000 people in 2010 and 190 mg/day/1000 people in 2011 at an Australian music festival. Similarly, [53] ([53]) reported median MDMA consumption estimates of 21,030 mg consumed/d/1000 in 2021 and 12,660 mg consumed/d/1000 in 2022, demonstrating both the substance’s popularity and temporal variations between festival years. [39] ([39]) observed significant increases in MDMA consumption during festivals in Slovakia and Czech Republic. [8] ([8]) documented a festival-related increase in MDMA during the Street Parade in Zurich, with influent concentrations rising from <20 ng/L during baseline weekdays to approximately 100 ng/L during the festival event, representing a >5-fold increase. [9] ([9]) reported MDMA as the most prominent substance across six European festivals, while [23] ([23]) detected MDMA consumption ranging from 18 after the event to 613 ng/L during the Carnatal festival in Brazil. The consistency of MDMA detection across diverse analytical approaches, festival types, and international contexts underscores its role as a primary substance of choice at music festivals.

Cannabis, measured through its main metabolite THC-COOH (Tetrahydrocannabinol), represented the second most frequently detected substance category. [9] ([9]) documented THC-COOH presence in all pooled urine samples from six European festivals, while [36] ([36]) consistently detected cannabis metabolites across all sampling days at an Australian festival, similar to [13] ([13]), who identified cannabis derivatives including THC-COOH across multiple New South Wales festivals, and [8] ([8]), who documented THC-COOH presence primarily during festival periods. [53] ([53]) reported median cannabis consumption ranging from 3145 to 4234 mg consumed/d/1000 people.

Cocaine consumption, primarily measured through its metabolite BZE, demonstrated variable, but consistent detection patterns. [36] ([36]) reported cocaine loads of approximately 3.5 mg/day/1000 people for cocaine, but an average between 37 and 45 mg/day/1000 people for BZE across both festival years. [38] ([38]) reported cocaine detection frequencies of 28% across sampling campaigns. [23] ([23]) documented cocaine concentrations ranging from 760 after the event to 8914 ng/L during the Brazilian Carnatal festival, while [9] ([9]) detected cocaine across multiple European festival sites. [13] ([13]) found cocaine detection frequencies of varying levels across Australian festivals, indicating substantial use patterns despite generally lower detection frequencies compared to MDMA and cannabis.

Amphetamine and methamphetamine patterns varied significantly across geographical regions, reflecting regional consumption preferences rather than universal festival trends. [39] ([39]) identified methamphetamine as historically prevalent in Czech and Slovak populations, resulting in minimal festival-specific consumption changes despite high baseline concentrations. Conversely, amphetamine showed more pronounced festival-associated increases across multiple studies, with [9] ([9]) and [13] ([13]) documenting detection frequencies that demonstrated clearer temporal relationships with event schedules.

Ketamine demonstrated emerging prominence as a festival-associated substance across multiple international studies. [9] ([9]) noted increased ketamine concentrations in 2016 samples compared to previous years, suggesting evolving consumption trends. [53] ([53]) reported substantial ketamine consumption estimates ranging from 15,700 to 18,430 mg consumed/d/1000, while [13] ([13]) consistently detected ketamine across multiple Australian festivals. [30] ([30]) documented ketamine as one of the highest-concentration substances detected during their festival study (138,000 ng/L), indicating significant user adoption globally.

NPS appeared with lower frequency and concentrations compared to traditional drugs across all studies. [12] ([12]) detected five NPS at concentrations ranging from 0.4 to 8.6 ng/L, primarily synthetic cathinones including mephedrone and methcathinone. [9] ([9]) identified 21 different NPS across six European festivals, with synthetic cathinones and phenethylamines being most detected. [5] ([5]) successfully identified α-D2PV (Novel synthetic cathinone, part of the broader class of NPS) and its metabolites, demonstrating analytical feasibility while confirming relatively limited prevalence. [13] ([13]) detected higher than expected levels of cathinones in wastewater that were not reported by survey respondents, highlighting the utility of wastewater analysis for detecting unexpected adulterants. The low detection frequencies and concentrations suggest that while NPS are present in festival environments, they represent supplementary rather than primary consumption choices.

Prescription and pharmaceutical substances demonstrated consistent background presence across festival samples. [39] ([39]) included pharmaceuticals in their screening approach, with little observed increase in tramadol, citalopram, venlafaxine, codeine, and oxazepam, while [11] ([11]) screened for 120+ pharmaceuticals, finding high enough concentrations for only a few (e.g., tramadol, venlafaxine). [30] ([30]) used separate methods for pharmaceuticals and recreational drugs, finding that, out of a total of 13,950 g of emerging substances, 20.8% were analgesics, 5.7% antibiotics, and 2.7% other pharmaceuticals. [53] ([53]) reported detection of substances including benzodiazepines (temazepam, diazepam), opioids (methadone, codeine, tramadol), and stimulants (pseudoephedrine, ephedrine), indication diverse pharmaceutical substance exposure in festival populations.

Polydrug use patterns emerged from multiple validation approaches. [22] ([22]) employed triangulation approaches combining self-reported data with oral fluid and pooled urine samples to confirm polydrug consumption. [13] ([13]) identified substances in wastewater that were not reported in surveys, indicating complex consumption behaviours involving both intended and unintended substance use. These findings align with wastewater detection of multiple substance categories within individual samples, suggesting complex consumption behaviours rather than single-substance preferences.

#### 3.3.4. Music Genre–Drug Associations

Several studies revealed distinct patterns of substance use that appeared to correlate with specific music genres and festival types. The most prominent association identified was between EDM events and elevated consumption of MDMA. [22] ([22]) performed a comparative analysis of three Norwegian festivals and discovered that a markedly higher percentage of participants at the EDM event reported MDMA usage (3.9%) relative to attendees at other music festivals (0.7%). This finding was supported by wastewater analysis data showing elevated MDMA concentrations during EDM events compared to other music genres, where cocaine, THC-COOH, and NPSs prevailed. However, [36] ([36]) noted that “MDMA use is not limited to the electronic/techno dance scenes” as these specific music genres were not being represented at the festival that they covered.

[39] ([39]) examined wastewater from music festivals across the Czech Republic and Slovakia, covering multiple genres including with over 130,000 active festival attendees. The study demonstrated that MDMA and cocaine were primarily consumed during dance, pop, and multi-genre festivals, while cannabis use showed increases during pop/rock festivals. Notably, metal and country/folk festivals showed no significant changes in drug consumption patterns. The study’s comprehensive approach included previous research by [40] ([40]) which had identified cocaine and codeine, as well as methamphetamine, as dominant substances at country/folk festivals.

#### 3.3.5. Regional Patterns in Festival Drug Consumption

A comprehensive analysis of studies across multiple continents revealed regional variations in drug consumption patterns at music festivals (see Table 3), reflecting distinct geographical differences shaped by historical, socio-economic, and cultural factors.

Central and Eastern European countries exhibited a methamphetamine-dominant pattern rooted in historical factors. [39] ([39]) extended their previous analysis across both Slovakia and the Czech Republic, revealing that “the EMCDDA counts Slovakia and the Czech Republic among countries with dominant methamphetamine consumption”, rooted in the easy and cheap production of methamphetamine during the Iron Curtain period. Additional studies by [11] ([11]) and [12] ([12]) further confirmed this regional pattern in Slovakia, while documenting geographic variations within the country, with highest cocaine concentrations in Bratislava and highest methamphetamine in rural areas. Croatian studies by [58] ([58]) identified 46 total substances (26 classic drugs plus 20 NPS) from pooled urine samples, while [56] ([56]) documented a 30-fold MDMA increase during festivals. Serbia, included in the six-country study by [9] ([9]), demonstrated MDMA-centric festival culture with clear dominance over other substances, representing some of the highest festival-associated increases documented globally. Hungarian festival research by [38] ([38]) identified environmental contamination of lake ecosystems with psychoactive substances, including MDMA, cocaine, and others, as well as methamphetamine, keeping in line with the central and eastern European trend.

Western European countries generally exhibited MDMA-dominant festival patterns with notable sub-regional variations. Belgian studies by [9] ([9]), [21] ([21]), and [35] ([35]) revealed MDMA as having the highest concentrations among traditional drugs, with critical safety concerns regarding dangerous cardiac medications (flecainide and amlodipine) detected as cocaine adulterants. Spanish research by [10] ([10], [9]) demonstrated significant MDMA increases during festival periods. Switzerland presented interesting time patterns, with [7] ([7]) conducting two-year festival monitoring (2014–2015), showing that MDMA and amphetamine increased significantly per capita during festivals while cocaine remained stable, attributed to population-increase effects only. [8] ([8]) documented acute drug loading impacts following the 2009 Street Parade in Zurich, revealing dramatic post-event concentration spikes of cocaine, BZE, and amphetamine-like stimulants (amphetamine and MDMA) in influent wastewater compared to reference weekday samples, with the authors noting that heavy rainfall during the event likely diluted samples and actual concentrations would have been even higher under dry conditions. French studies by [15] ([15]) found no specific festival effect, a result that they attributed to the public nature and diverse audience of the monitored event. [9] ([9]) also included Portugal, revealing cannabis with the highest increase (10-fold) during festival periods. The United Kingdom showed a large-scale in NPS diversity, with [35] ([35]) documenting 53 different NPS detected. [9] ([9]) included the UK in their six-country study, revealing temporal patterns with more NPS detected in 2015 versus 2016, while ketamine concentrations increased significantly in 2016, potentially indicating the country’s role as a global pioneer in emerging substance monitoring.

When it comes to Scandinavian countries, Danish research by [26] ([26]) stated that ketamine was detected using UPLC-HR-TOF-MS. Norwegian festival drug consumption patterns present conflicting evidence regarding amphetamine-type stimulants. [9] ([9]) found that Norway was unique among European countries studied, as amphetamine and methamphetamine levels in pooled urine samples were “only sufficiently high in Norway to allow accurate quantification”. However, this finding directly contradicts [22] ([22]), who conducted individual testing of 651 Norwegian festival attendees and found 0% tested positive for amphetamine. These contradictory findings suggest either methodological differences between pooled versus individual sampling approaches, or potentially different populations/time periods being studied. The most commonly detected substances in Norwegian festivals were cannabis, MDMA, and cocaine, consistent with broader European patterns.

Australian studies consistently demonstrated high MDMA and cannabis combinations, establishing Australia as the first country to apply design of experiments (DOE) optimisation for detection of 71 NPS. Australian research revealed clear festival versus residential differences, with [36] ([36]) specifically noting that “MDMA consumption at the festival was substantially higher than in the urban area (4–58 vs. 1–7)”, demonstrating the festival-specific elevation effect characteristic of the region.

Taiwanese research by [30] ([30]) focused on environmental monitoring during the Spring Scream Festival, analysing 30 sampling sites, with concentrations of ketamine and MDMA peaking during festival periods, representing a pollution control perspective rather than epidemiological focus.

[23] ([23])’s study in Brazil revealed a cocaine-dominant pattern, unusual compared to global MDMA dominance at music festivals. Particularly significant was the detection of cocaethylene (COE), where “COE metabolite commonly produced by individuals after the concomitant use of cocaine and alcohol, was detected only in samples collected during the Carnaval”, demonstrating region-specific polydrug use patterns combining cultural drinking practices with local drug availability. 

Table 4 provides a brief summary of the main findings.

## 4. Discussion

### 4.1. Summary of Results

To our knowledge, this is the first comprehensive systematic review and narrative synthesis to thoroughly examine wastewater analysis methods for detecting psychoactive recreational substances at music festivals worldwide. This review includes 22 studies, spanning festival monitoring across multiple continents and reflecting diverse methodological approaches to WBE. The studies encompass festivals with widely varying attendance, from approximately 6200 to 600,000 attendees, that were published between 2013 and 2024, reflecting the emerging and rapidly evolving field of festival-based substance monitoring. Where demographic data were reported, festival attendees showed varied age profiles, with some studies documenting broad age ranges spanning from adolescents to seniors ([53]), while others found more specific demographic segments depending on festival type and genre ([22]; [36]).

#### 4.1.1. Sampling Methodologies and Analytical Approaches

The methodological approaches employed across the included studies reflect both the evolving sophistication of WBE and the unique challenges presented by festival environments. The predominance of 24 h composite wastewater sampling strategies suggests that researchers have prioritised population-level assessment over real-time monitoring capabilities, though this preference may reflect infrastructural constraints rather than optimal research design. Sampling frequency can only be considered in a meaningful way when appropriate sewer infrastructure is in place. In urinals, portable toilet blocks or portaloos, drug residue stability is the decisive factor. Once excreted, the sample remains in the collecting container until the unit is emptied—typically after the event or at night.

The strategic choice between wastewater and pooled urine matrices across studies reveals fundamental trade-offs between analytical sensitivity and population representativeness. While pooled urine sampling approaches ([9]; [13]; [22]; [35]; [58]) offered superior analyte concentrations and reduced dilution effects, enabling enhanced detection of NPS, the demographic limitations of these approaches highlight significant representativeness concerns. The successful detection of 77 drugs and metabolites through UPLC-HR-TOF-MS compared to limited immunoassay detection in pooled urine ([26]) definitively established the inadequacy of simpler screening approaches, influencing subsequent methodological choices across the field.

The analytical evolution from targeted LC-MS/MS approaches toward HRMS techniques ([5]; [9]) reflects the field’s adaptive response to rapidly changing recreational drug markets. However, the observation by [35] ([35]) regarding the lack of commercial standards for most metabolites highlights the challenge in current analytical capabilities that extends beyond instrumental sophistication to encompass broader knowledge gaps in NPS metabolism and excretion patterns.

The widespread adoption of SPE across 17 studies represents methodological standardisation driven by practical necessity rather than theoretical optimisation. However, alternative approaches to traditional SPE protocols have demonstrated viability in specific contexts.

[41] ([41]) suggested that direct injection methods can achieve comparable sensitivity and that this might have implications for the scalability and accessibility of festival monitoring programmes, particularly in resource-limited settings where sophisticated sample preparation infrastructure may not be available. However, the frugality of direct injection methods is debatable, as the equipment used for direct injection is cost-prohibitive for most laboratories and requires constant maintenance to maintain sensitivity. SPE manifolds, in contrast, are comparably cheap and can be operated by a person with minimal training. The mass spectrometer, however, requires specialist operators and resources.

#### 4.1.2. Substance Detection Patterns, Geographic Variations, and Festival Characteristics

The substance detection patterns revealed across festival settings demonstrate both consistent global preferences and profound regional variations that challenge simplistic assumptions about recreational drug use in festival environments. The dominance of MDMA across diverse geographical contexts establishes this substance as a defining characteristic of contemporary festival culture. However, the substantial variations in consumption estimates, ranging from [53] ([53]) with median estimates of 12,660–21,030 mg consumed/d/1000 people to more modest levels in other studies, suggest that festival-specific factors, analytical methodologies, or regional preferences significantly influence detected patterns.

The geographic variations in substance preferences reveal underlying social, economic, and cultural factors that extend far beyond festival characteristics. The observation by [39] ([39]) that “the EMCDDA counts Slovakia and the Czech Republic among countries with dominant methamphetamine and cannabis consumptions”, rooted in historical methamphetamine production during the Iron Curtain period, demonstrates how political and economic transitions shape contemporary recreational drug markets. Similarly, the cocaine-dominant pattern identified by [23] ([23]) in Brazilian festivals, contrasting sharply with global MDMA predominance, reflects regional drug availability and cultural consumption patterns that transcend festival-specific influences. The unprecedented diversity of NPS detected in UK festivals compared to Belgian and Norwegian events ([9]; [35]) suggests that drug market dynamics vary significantly even within relatively homogeneous European contexts, challenging the development of standardised monitoring protocols.

Central and Eastern European methamphetamine dominance, South American cocaine prevalence, Western European MDMA-centricity, and unique and contradicting Scandinavian patterns each reflect distinct historical, economic, and cultural influences on regional drug markets. The detection of region-specific preferences demonstrates that standardised international monitoring protocols may miss critical regional variations. Understanding these geographical differences is essential for developing culturally appropriate, pharmacologically relevant festival safety initiatives and tailored evidence-based harm reduction strategies.

The influence of music genre on substance use patterns represents an additional layer of complexity that intersects with but transcends geographical variations. The strong association between EDM events and MDMA consumption, demonstrated by [22] ([22]), suggests that musical programming significantly influences drug preferences beyond regional availability patterns. [39] ([39]) provided comprehensive evidence of genre-specific consumption patterns, documenting that MDMA and cocaine were primarily consumed during dance, pop, and multi-genre festivals, while cannabis use increased during pop/rock festivals, and notably, metal and country/folk festivals showed no significant changes in drug consumption patterns. However, the observation by [36] ([36]) that “MDMA use is not limited to the electronica/techno dance scenes” indicates that while genre associations exist, they may not be deterministic, with festival culture and attendee expectations potentially mediating these relationships. As only some of the analysed studies compared specific music genres or included mixed genres, definitive conclusions regarding the association between music genres and consumed drugs should not be made. Preliminary conclusions may introduce bias, reinforcing misleading associations.

Temporal consumption patterns revealed consistent increases during festival periods compared to baseline measurements across multiple studies. The capacity of large-scale festivals to generate acute loading events on wastewater infrastructure was demonstrated by [8] ([8]), by comparing influent samples collected on a reference day with those obtained immediately after the event, which revealed significant concentration spikes, with elevated cocaine, BZE, and amphetamine-like stimulants (amphetamine and MDMA). Critically, the authors noted that heavy rainfall during the event likely diluted these measurements, suggesting that actual substance loads under dry conditions would have been substantially higher. [23] ([23]) demonstrated significant festival-to-baseline ratios for key substances, with MDMA and cocaine showing the most pronounced increases during event periods. [39] ([39]) observed similar temporal patterns across seven different festivals with varying musical genres, while [9] ([9]) documented consistent day-to-day increases throughout festival periods. These temporal dynamics suggest that festival environments fundamentally alter consumption behaviours, with important implications for harm reduction timing and medical resource allocation strategies.

#### 4.1.3. Public Health and Harm Reduction Applications

The application of wastewater monitoring for public health and harm reduction purposes represents one of the most promising yet underdeveloped aspects of festival-based epidemiology. The capacity for population-level, anonymous monitoring demonstrated across studies provides unique opportunities for evidence-based intervention design, yet the translation of analytical findings into effective harm reduction responses remains challenging. [13] ([13]) emphasised the potential for wastewater analysis to be “triangulated with complementary data sources currently available in Australia such as police seizures, surveys, medical presentations, information from peer-based harm reduction services and toxicological analyses to feed into public health alerts”, highlighting both the integrative potential and current fragmentation of festival surveillance systems.

The detection of dangerous adulterants and NPS through WBE offers critical safety benefits that extend beyond traditional surveillance capabilities. The identification of cardiac medications such as flecainide and amlodipine as cocaine adulterants ([21]) and the detection of highly toxic substances such as N-ethylpentylone and norfentanyl ([13]) demonstrate the capacity of wastewater monitoring to identify specific health risks that might otherwise remain undetected until clinical presentations occur. However, the temporal lag between sample collection, analysis, and result availability fundamentally limits the utility of current approaches for immediate harm reduction responses during ongoing events.

The pioneering detection of emerging substances, including the world’s first identification of kavain (the metabolite of kava) and psilocin (the metabolite of psilocybin or “magic mushrooms”) in festival wastewater ([53]), illustrates the potential for wastewater monitoring to serve as an early warning system for NPS entering recreational drug markets. This capability is particularly valuable given the rapid evolution of NPS markets and the challenges faced by traditional surveillance methods in detecting emerging substances. However, the successful implementation of early warning systems requires sophisticated data interpretation frameworks and rapid communication channels that currently remain underdeveloped across most festival contexts.

The integration of wastewater monitoring with complementary surveillance approaches, as demonstrated by [13] ([13]), [22] ([22]), and [53] ([53]) through triangulation of surveys, oral fluid samples and wastewater/pooled urine data, reveals the potential for comprehensive monitoring systems that overcome individual methodological limitations. The demonstrated capacity for objective, population-level surveillance provides a foundation for evidence-based harm reduction approaches, yet successful implementation depends on addressing current temporal, technological, and coordination limitations through sustained methodological development and stakeholder engagement.

While this review focuses on festival WBE as a public health tool, it is important to acknowledge that the detected substances also pose environmental risks. Studies like [38] ([38]) as well as [1] ([1]) have demonstrated that festivals can transiently contaminate aquatic ecosystems with biologically active compounds that retain pharmacological activity and may adversely affect wildlife.

Understanding acute loading patterns has implications beyond immediate harm reduction, extending to wastewater treatment plant operations and environmental monitoring. The observation by [8] ([8]) that sewage treatment plants can be exposed to brief periods of exceptionally high drug concentrations during major festivals raises questions about the capacity of standard treatment processes to handle such acute loads, and whether these events contribute disproportionately to downstream environmental contamination. This infrastructure perspective complements the individual-level harm reduction focus of most festival monitoring studies, highlighting the broader environmental and operational implications of mass gathering substance use.

Stakeholders from all involved groups (e.g., festival goers, festival organisers, musicians, political and cultural representatives, physical and mental health experts, WBE experts and environmental associations) should discuss and develop actionable recommendations for public health surveillance, harm reduction and festival interventions.

### 4.2. Limitations

This systematic review identified several methodological constraints and limitations that affect the interpretation and generalisability of wastewater-based epidemiological studies conducted at music festivals. These limitations span multiple domains, including study design, analytical methodology, sampling constraints, and data interpretation challenges. In addition, our systematic review has its own limitations.

#### 4.2.1. Limitations of This Systematic Review

The aim of this systematic review was to evaluate and synthesise the existing literature on recreational drug consumption at music festivals based on wastewater analysis. Thus, it had an aim rather than a specific research question.

We restricted the review to wastewater analysis. Therefore, we did not capture many important aspects of substance use at music festivals. Wastewater analysis measures the consumed substances of the group or a subgroup of festival goers, not the drug consumption of one individual. Moreover, as drugs can be contaminated, individuals might believe they are consuming different substances than those detected. To investigate such discrepancies, mixed-method studies including individual qualitative and quantitative interviews as well as individual drug tests could provide valuable contextual insights.

Based on this review, future researchers might develop more specific questions, for example, what the specific differences in drug and alcohol abuse between electronic and folk music festivals are. Such a more specific review should follow the PICO (population, intervention, comparison, outcome) approach to formulate a research question.

We only included studies published in English, German, and Romanian. Publications in other languages might have been missed.

Our literature search was restricted to PubMed, MEDLINE and Embase. The decision was made because these databases focus on biomedical science, medicine, drug research and pharmacology. However, databases focused on psychological research such as APA PsycInfo might contain additional relevant articles.

#### 4.2.2. Geographic Representation and Global Applicability

A primary limitation of this review is that the current literature exhibits significant geographic concentration, with studies predominantly conducted in European countries. Notable gaps exist in coverage from major regions including the United States, Africa, and large portions of Asia and South America. This geographic bias fundamentally limits the global applicability of findings. The concentration of research in higher-income countries with established wastewater infrastructure may not adequately represent patterns in regions with different socioeconomic contexts or emerging festival markets.

#### 4.2.3. Demographic Representation, Epidemiology and Gender Bias

Festival events create complex population dynamics that complicate data interpretation. [56] ([56]) note that the intake of psychoactive substances seen during the festival week cannot be deemed indicative of permanent residents. This also ties in with the impossibility of determining exact contributor numbers. [35] ([35]) acknowledge that population fluctuations within festival events cannot be predicted, making accurate normalisation of consumption estimates problematic. Furthermore, wastewater sampling at festivals cannot determine how many people consumed recreational drugs and how many of these had a substance use disorder.

Another pervasive limitation across multiple studies is the systematic demographic bias toward male populations in pooled urine sampling approaches. [26] ([26]) explicitly acknowledge that the urinals were designed exclusively for men, as “the design of the urinals did not favor easy use for females”. [21] ([21]) explain that technical requirements for sample processing prevent the inclusion of mixed waste, effectively limiting data collection to male urinal users only. This systematic exclusion creates substantial gaps in understanding substance use patterns among female festival attendees and may miss important gender-specific differences in drug preferences, consumption patterns, and associated health risks. However, female underrepresentation and the lack of knowledge on gender differences can be overcome in the future with suitable infrastructure and technical adaptations.

#### 4.2.4. Sampling Challenges

Festival-based sampling presents unique coordination complexities constrained by existing infrastructure and technical limitations that significantly impact study feasibility and introduce potential selection bias. Research groups document the substantial communication requirements between festival organisers, volunteers, and wastewater treatment operators, with success heavily dependent on stakeholder willingness to participate ([9]). [13] ([13]) describe how practical constraints related to the number of available facilities and limited sampling windows create additional barriers. These coordination requirements may result in convenience sampling of more cooperative festival organisers rather than representative sampling across diverse festival types, sizes, and management approaches. International research coordination introduces additional bureaucratic complexities, potentially limiting cross-national comparative studies.

Environmental conditions and chemical contamination present ongoing challenges to sample integrity and analytical accuracy. Multiple studies document weather-related interference, with researchers reporting contamination from dilution by non-urine liquids and variable exposure to rain at uncovered sampling sites ([26]; [21]). [38] ([38]) highlight the difficulty in determining contamination sources, noting the uncertainty about whether detected substances derive from direct festival use or inappropriate wastewater drainage from surrounding areas.

Furthermore, sample integrity during collection and storage presents ongoing analytical accuracy challenges. [35] ([35]) warn that “there is a chance of missing unstable targets during analysis if sample collection and storage (freezing) are delayed”. These storage and stability issues may be particularly problematic in festival settings where immediate laboratory processing is impractical. Additionally, the stability of target molecules might be impacted by contamination of the samples with sanitiser or deodoriser.

#### 4.2.5. Analytical Methodology and Technical Constraints

Another potentially critical aspect is the limited availability of commercial reference standards for NPSs. [35] ([35]) highlight that “most metabolites are not available as commercial standards so to be included in targeted methods”. Similarly, multiple studies describe the rapidly changing nature of the NPS market as a fundamental constraint when applying standard analytical techniques ([13]; [26]; [41]). This limitation restricts researchers to qualitative identification methods for many substances and prevents accurate quantification of emerging drug classes.

Complex sample matrices present significant analytical challenges across multiple studies. [41] ([41]) identify them as one of the primary challenges in LC-MS analysis, finding that “only 75% of tested compounds (53 out of 71 NPS) achieved acceptable matrix effects”. The absence of standardised analytical protocols creates significant comparability challenges across studies. [26] ([26]) document substantial methodological inconsistencies that exist between the outcomes of bioanalytical tests and drug-screening immunoassays, with the latter yielding both false negative and false positive results. Such variation in detection thresholds and analytical approaches can lead to dramatically different conclusions about substance prevalence and may compromise meta-analytical approaches.

#### 4.2.6. Data Interpretation Challenges

Pooled sampling approaches are another potential confounder, with both [9] ([9]) and [35] ([35]) acknowledging that pooled urine data represents a momentary capture of substance use rather than comprehensive temporal patterns. This limitation prevents researchers from capturing dynamic consumption patterns throughout festival events and may miss important variations in substance use intensity, timing, and sequential polydrug use patterns.

In addition, the absence of historical baseline data complicates trend identification and context interpretation. [23] ([23]) recommend comparing metabolite ratio values from suspected dumping events with historical data from the same location, while noting that their study represents the first wastewater monitoring research conducted in Natal. The general absence of longitudinal data limits researchers’ ability to distinguish festival-specific consumption patterns from broader temporal trends in substance use.

Differential excretion profiles across substance classes create interpretation challenges for temporal analysis. [9] ([9]) note that substances exhibit varying excretion rates; for instance, cannabis has significantly prolonged excretion rates compared to drugs such as MDMA. [22] ([22]) highlight additional complexity related to extensive metabolism of many NPS, which results in predominantly metabolite detection with minimal parent drug concentrations in urine samples. These pharmacokinetic differences mean that detection patterns may not accurately reflect contemporaneous consumption patterns, potentially leading to misinterpretation of substance use timing and relative consumption intensities.

Another point to consider is the rate of drug clearance from the body. During a festival event spanning less than a day, there is a high chance that drugs used during the festival may only be metabolised and excreted after the event, at a different site. It is also possible that drugs excreted at the site were consumed well prior to the event. Drug loads in wastewater from festivals should therefore be interpreted with caution.

Wastewater analysis uses pooled specimens and can only provide a positive test of a drug or the concentration of a drug within the whole sample, not for the individual person. This method is unable to ascertain the proportion of festival goers who consume a substance. A small subgroup of festival visitors or even a single visitor can influence the overall result. Thus, data derived from wastewater analysis at music festivals, as well as secondary research studies based on such analyses, must take this methodological shortcoming into account to avoid conclusions about individual festival participants or the totality of festival goers.

#### 4.2.7. Study Design and Cross-Study Comparability Issues

Methodological heterogeneity across studies creates substantial challenges for evidence synthesis and meta-analysis. [9] ([9]) caution that data interpretation between different festivals and countries requires careful consideration due to variations in festival characteristics, populations, timeframes, sample matrices, and collection methods. This methodological heterogeneity fundamentally limits the ability to conduct robust comparative analyses and develop generalisable conclusions.

#### 4.2.8. Risk of Bias

We already mentioned potential biases regarding epidemiology and gender above. However, the design of the included studies did not involve randomisation, concealment, specific care or treatment. Therefore, selection and performance bias are not relevant for this systematic review. Detection bias is possible, as only pre-selected drugs were measured and most metabolites are not available as commercial standards. Attrition bias or loss to follow-up bias does not apply, because the studies were single-timepoint studies.

#### 4.2.9. Causes of Heterogeneity

Prior to this systematic review, the evidence for assumptions regarding the causes of heterogeneity was scarce. However, given the obtained results, geographic and demographic conditions, the music genre, sampling and analytical methods might be considered as causes of heterogeneity in future cross-sectional, epidemiological and cohort studies using wastewater analysis for psychoactive substances at music festivals. Given the nature of the obtained data, a sensitivity analysis to assess the robustness of the synthesised results was not deemed meaningful.

### 4.3. Implications and Future Directions

This review supports the significant potential of festival WBE as an objective, population-level tool for substance use monitoring, despite substantial methodological limitations that currently constrain its reliability and global applicability.

Given the identified limitations, several critical recommendations for future research are proposed.

First, there is an urgent need to develop standardised protocols for sampling, analysis, and data reporting that can be implemented consistently across different geographical and cultural contexts. The current methodological heterogeneity prevents meaningful cross-study comparisons and meta-analysis, fundamentally limiting the field’s ability to build a robust evidence base. Such protocols should consider that portaloos also collect sanitisers and deodorisers which might impact drug residue stability. Establishing international consensus guidelines would enable systematic evidence synthesis and accelerate knowledge accumulation. Currently, WBE cannot measure polydrug use which has the strongest association with harm. As festival testing is mostly done after the event, it can only inform the planning for the next, but usually not the medical and psychological support for an ongoing event.

Second, innovative sampling approaches must be developed to address the systematic demographic bias inherent in current pooled urine methodologies. The exclusion of approximately 50% of festival populations in male-only sampling approaches fundamentally compromises the validity and public health relevance of findings.

Third, comprehensive efforts should be undertaken to expand reference standard availability for emerging psychoactive substances. The rapid evolution of NPS markets requires analytical capabilities that extend beyond currently available commercial standards. International collaboration is needed to establish coordinated synthesis programmes, rapid response protocols for novel substances, and predictive modelling approaches for metabolite identification.

Fourth, research should focus on developing and validating simplified analytical approaches that maintain reliability while reducing resource requirements and improving global accessibility. The demonstration that direct injection methods can achieve comparable sensitivity for specific applications suggests significant opportunities for making festival monitoring more feasible for resource-limited settings.

Fifth, longitudinal studies with multi-year follow-up periods should be conducted to distinguish festival-specific consumption patterns from broader temporal trends in recreational drug use. Current cross-sectional approaches provide limited insights into the stability and evolution of substance use patterns, hampering the development of evidence-based interventions and policy responses.

Sixth, future research on genre-specific differences risks stigmatising attendees without clear public health benefit. Future studies should therefore prioritise broader patterns of use and contextual factors rather than simply contrasting genres. In this respect, we would like to mention that drug and alcohol consumption are commonplace at various types of festivals, not only those with a focus on music. [6] ([6]), for example, collected wastewater over several consecutive weeks from a location where a school-leaver festival was held and found significant changes in MDMA and MDA concentrations over the festival week.

Seventh, comprehensive research should focus on the environmental impact of detected substances on aquatic ecosystems and microorganisms. Research demonstrated that festivals may transiently contaminate lake ecosystems with biologically active substances, with most recreational drugs retaining their pharmacological activities and potentially causing adverse impacts on wildlife. Additionally, monitoring studies have detected drug residues in lakes and rivers receiving wastewater effluents from festival catchment areas, highlighting the downstream environmental implications of acute loading events. Given these findings, systematic ecological risk assessments are urgently needed to understand long-term ecosystem consequences and develop mitigation strategies for environmental protection.

Finally, comprehensive ethical and privacy frameworks must be established for population-level substance monitoring. While wastewater epidemiology offers significant public health benefits, concerns about privacy, potential stigmatisation, and misuse of data for punitive rather than health-promoting purposes require careful consideration and regulatory oversight. However, there is significant public health interest to protect young people from drug use. One could argue that this interest can be seen as a higher ethical priority than protecting people’s individual privacy. In fact, wastewater analysis appears as a feasible compromise to gain knowledge without identifiable individual data.

Addressing these research priorities will be crucial to establishing festival wastewater epidemiology as a reliable, evidence-based surveillance tool. If successfully implemented, these advances could enable the development of sophisticated early warning systems, evidence-based harm reduction interventions, optimised medical resource allocation, and international cooperation frameworks for addressing emerging substance markets.

## 5. Conclusions

This systematic review has provided the first comprehensive assessment of wastewater analysis methods for detecting psychoactive substances at music festivals worldwide. The findings establish WBE as a valuable, objective tool for monitoring substance use patterns at large-scale music events, offering significant advantages over traditional survey-based approaches through real-time detection capabilities and population-level assessment. The evidence demonstrates that both wastewater sampling from sewage systems and pooled urine collection approaches can effectively detect conventional substances and emerging psychoactive compounds. These methods provide critical insights for public health surveillance, emergency medical preparedness, and evidence-based policy development for large-scale events. The temporal patterns observed confirm festivals as high-risk environments, with consumption levels significantly exceeding background population rates. However, significant methodological limitations constrain the current evidence base. Geographic concentration of research limits global applicability, while systematic gender bias in pooled urine sampling approaches compromises population representativeness. Limited availability of reference standards for emerging substances restricts analytical capabilities, and methodological heterogeneity prevents robust comparative analyses across studies. Future research should prioritise developing standardised international protocols, gender-inclusive sampling methods, and enhanced analytical capabilities for NPS. Critical needs include simplified approaches that maintain reliability while improving global accessibility and expanded international coordination to establish consensus guidelines and rapid response capabilities. Despite these limitations, this review establishes wastewater analysis as a tool for objective substance use monitoring at music festivals. The evidence supports its integration into comprehensive public health surveillance systems and harm reduction strategies, offering event organisers and public health authorities objective data to guide intervention planning and resource allocation.

## Figures and Tables

**Figure 1 behavsci-15-01672-f001:**
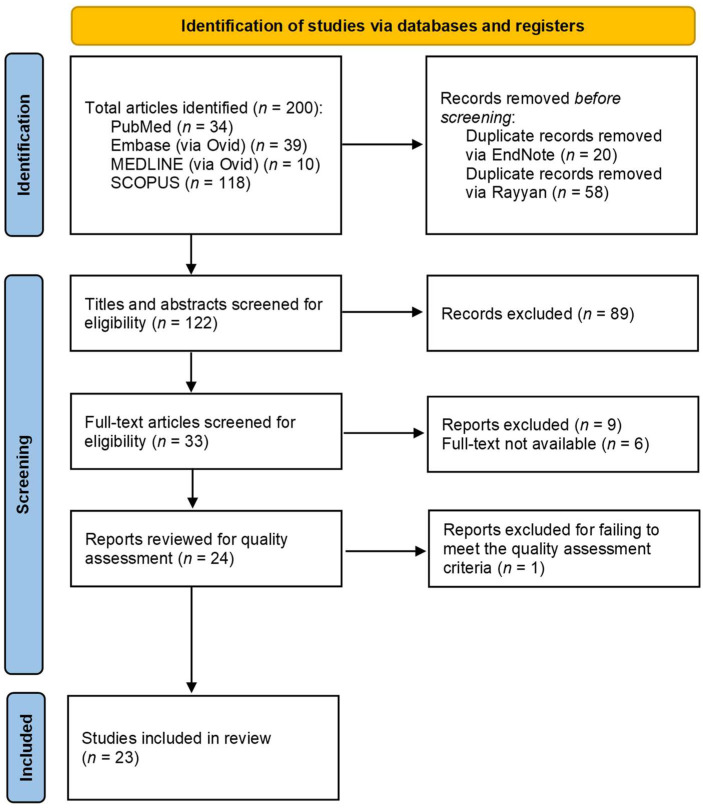
The PRISMA Flow Diagram of the selection process.

**Table 1 behavsci-15-01672-t001:** Detailed summary of included studies.

Author(s), Year, Country	Festival and Sample Details	Sampling Method	Analytical Technique	Substances Detected	Main Findings	Additional Notes
([5]), Australia	One multi-day music festival in Australia (May 2022); samples collected from an on-site treatment facility	24 h composite wastewater samples analysed using a targeted approach informed by in vitro metabolism assays	LC-MS/MS + LC-–HRMS supported by in vitro metabolism for confirmation of drug metabolites	α-D2PV (parent compound) and three metabolites (M1, M2: hydroxylated; M4: dehydrogenated)	First WBE application targeting α-D2PV; results demonstrate potential for integrating in vitro data to support NPS detection in wastewater	Qualitative detection only (no reference standards available); proposed method supports improved biomarker identification for future NPS monitoring
([7]), Switzerland	Two consecutive editions (2014 and 2015) of a multi-day music festival in Switzerland, attended by ~50,000 people daily; samples were collected from the local sewage treatment plant, which received all wastewater from the festival and the city	24 h composite wastewater samples collected during both festival and baseline periods	SPE-LC-MS/MS was used to quantify 14 target compounds; statistical modelling was applied to account for uncertainty	MDMA, cocaine, AMPH, MAMPH, BZE, THC-COOH, 6MAM; ketamine, mephedrone and methylone were detected but not quantified; LSD and BZP were not detected	MDMA and AMPH showed significantly higher per capita loads during the festival compared to baseline, suggesting increased consumption during the event; cocaine use remained stable; MAMPH increased slightly but remained low; findings were consistent across both years	Comparison with seizure data suggested that MDMA and AMPH were often purchased on-site, supporting the existence of a festival drug market; authors recommend targeted prevention measures such as drug checking
([8]), Switzerland	Street Parade 2009 in Zurich (600,000 participants); samples collected from five major street parades in Switzerland (Bern, Basel, Geneva, Lucerne, Zurich) from July to October 2009; comparative study conducted with reference sample versus sample collected day after the street parade	24 h flow rate-dependent composite wastewater samples; river water collected as grab samples (n = 22); lake water samples from Lake Thun, Lake Brienz, and Lake Biel at various depths	HPLC-MS/MS using large volume injection (100 μL) with minimal sample preparation; 12 drugs analysed	In influents and effluents: Cocaine, BZE, morphine, methadone and its main metabolite EDDP; followed by codeine and amphetamine; THC-COOH detected <LOQ; in rivers and lakes: BZE, methadone, EDDP most frequently detected	Major substances in influents: Cocaine, BZE, morphine (up to 2 μg/L), followed by codeine, methadone, EDDP (up to 600 ng/L), amphetamines and amphetamine-like stimulants (e.g., MDMA) (up to 100 ng/L); Cocaine, BZE, and amphetamine-like stimulants (e.g., MDMA) higher on day after event compared to reference sample	Street parades are associated with short-term concentration peaks
([10]), Spain	Large pop/rock music festival in Benicàssim, Spain (July 2008, ~40,000 attendees); samples collected daily from three STPs (Benicàssim, Castellón, Burriana) during three separate weeks: June (summer), July (festival), and January/April (non-festival)	Daily 24 h composite samples taken from influent and effluent wastewater; comparison made across seasonal and festival timepoints	SPE–UHPLC–MS/MS targeting 11 parent drugs and metabolites	Cocaine, BZE, MDMA, MDA, MDEA, amphetamine, methamphetamine, THC-COOH, BZE, cocaethylene; some seasonal or festival-specific variations	Significant increase in MDMA and MDA during the festival (MDMA >27 μg/L); cocaine and BZE prevalent across all sites; THC-COOH only detected in July	Results support MDMA being festival-associated
([9]), Multi-country (Europe)	Six music festivals in six European countries (UK, Belgium, Norway, Portugal, Serbia, Spain) were sampled between 2015 and 2018 using pooled urine (UK, BE, NO) and wastewater (PT, RS, ES)	Pooled urine from male urinals and 24 h composite wastewater samples at WWTPs; samples collected during peak festival hours or daily over event duration, as well as during a non-festival period	SPE + LC-MS/MS + LC-HRMS for targeted quantification of 197 NPSs, 6 illicit drugs and metabolites	Multiple illicit drugs and 21 NPSs identified; MDMA and cocaine (highest in UK and Belgium, Serbia and Spain), amphetamine and methamphetamine (Norway), THC-COOH (Portugal and Spain), ketamine and NPSs such as 4-FMC, 4-MEC, α-PVP, butylone, ethylone and MDPV (mainly in the UK)	Observed country-level preferences of illicit drugs; UK showed highest NPS diversity; ketamine concentrations increased in UK 2016 samples	First comprehensive comparative study across multiple European festivals; highlighted the demographic limitation of pooled urine sampling (male urinals); proposed combined use of wastewater and urine samples for improved representativeness
([11]), Slovakia	National study covering 15 Slovak cities (2013–2014); includes cities with music festivals such as Trenčín and Piešťany	24 h composite influent samples collected using automated samplers across 15 WWTPs; sampling covered weekdays and weekends; multiple samples per day were analysed	In-line SPE–LC-HRMS (Orbitrap)	Methamphetamine, MDMA, BZE, THC-COOH; also screened for psychiatric drugs and 120+ pharmaceuticals, but only the ones with the highest concentrations were discussed (e.g., tramadol, venlafaxine)	Methamphetamine was most commonly detected nationwide; MDMA concentrations peaked during festivals; cannabis use highest in Bratislava (112 mg/day/1000 inh) and Petržalka (74 mg/day/1000 inh); BZE levels suggested low cocaine use compared to Western Europe	Festival peaks observed in Trenčín (MDMA 207 ng/L) and Piešťany (159 ng/L); supports EMCDDA and police trends
([12]), Slovakia	Three music festivals in Slovakia: Pohoda (multicultural), Grape (dance), Lodenica (folk); ~50,000 attendees; samples from 9 cities over two years (2017–2018); 10 WWTPs	24 h composite influent samples collected every 15 min using automatic samplers; included samples before, during, and after festivals	SPE followed by LC-MS/MS for detection of 30 NPS (synthetic cathinones and phenethylamine)	EtS (for alcohol) and 5 NPSs including synthetic cathinones (mephedrone, methcathinone, buphedrone, pentedrone), and phenethylamines (25-iP-NBOMe)	Five NPS detected at low ng/L; mephedrone and methcathinone were most common	This was the first WBE study on NPS in Slovakia; confirmed higher NPS levels during festivals; mass loads support findings from other EU cities
([13]), Australia	Six single-day music festivals in New South Wales (NSW), Australia, held between March 2019 and March 2020; all were predominantly electronic dance music (EDM) events; festival attendance ranged from 6200 to 14,975	Wastewater sampled from portaloos and/or pooled tanks at end of event using pipette method	SPE followed by Ultra High-Performance Liquid Chromatography (UHPLC); screened for 98 psychoactive substances or metabolites	High-frequency detections: MDMA, MDA, cocaine, amphetamine, ketamine, methylone, alprazolam, diazepam, etizolam, oxazepam, temazepam; also detected: methcathinone, mephedrone, ethylone, N-ethylpentylone, norfentanyl	All major substances detected at all festivals; detection frequency for cathinones (e.g., mephedrone, methylone) higher than expected (suggesting adulteration or substitution); NPS not self-reported by survey participants	Wastewater sampling feasible and complementary to other data sources; limitations include inconsistent sampling and lack of quantitative analysis
([15]), France	World Music Day (June 21) in Bordeaux, monitored in 2017 and 2018; sampling from two WWTPs (~1 million inhabitants total)	24 h flow-proportional composite influent samples collected using refrigerated autosamplers; sampling occurred daily for one week including the Music Day	SPE + LC-MS/MS	cocaine, BZE, MDMA, THC-COOH, cocaethylene	Illicit drug use (cocaine, MDMA, cannabis) increased from 2017 to 2018; however, no specific increase was observed on World Music Day	Authors suggest lack of peer audience and public nature of event as reasons for no Music Day effect; first report of stable drug use during such a large event.
([21]), Belgium	Major international music festival in Flanders, Belgium, summer 2019 (~70,000 attendees per day); pooled urine samples collected at four different festival locations over three consecutive days (Friday–Sunday)	Pooled urine collected at men’s urinals via pumps; samples taken at 2 p.m., 5 p.m., 8 p.m., and 11 p.m.	GC-FID and LC-MS/MS (QTRAP) for 13 classical drugs; QTOF-MS for broad screening of NPS and adulterants	MDMA, MDA, ketamine, amphetamines, THC-COOH, cocaine (qualitative only), cocaethylene; NPS detected; also detected: flecainide, amlodipine, and other medications	MDMA and ketamine in all samples, often at high levels; cocaine confirmed in all samples qualitatively, but not quantitatively; two NPS (FMA and MMC) identified; widespread presence of prescription/OTC medications	Urine-based pooled sampling feasible for festival drug surveillance; limitations include male-only sampling and potential for environmental contamination
([22]), Norway	Three music festivals in Norway (EDM and pop/rock); two urban and one rural setting; ~651 self-report participants + pooled urine sampling from portable urinals	Self-reported surveys, oral fluid samples, and pooled urine from urinals; combined individual and anonymous data collection	Oral fluid: UHPLC-MS/MS; pooled urine: LC-MS/MS and LC-HRMS; in-house database used for NPS screening	THC-COOH, MDMA, cocaine, amphetamines, ketamine, and multiple NPS	Self-reports underestimated actual drug use; MDMA most frequently detected in both biological samples; NPS use confirmed by urine but not always self-reported; poly-drug use observed; cocaine and ketamine also common	Triangulation of biological and self-report data revealed discrepancies in reported vs. actual drug use
([23]), Brazil	Carnatal Festival, held in December 2022 Rio Grande do Norte, Brazil; samples collected across four days of the festival and the days before and after it	24 h composite influent wastewater samples collected from WWLS	SPE + UHPLC-MS/MS	BZE, cocaine, cocaethylene, MDMA detected throughout all days; BZE was most abundant	Cocaine use was substantially higher than MDMA across all days	First study to use WBE for a major music event in Brazil
([26]), Denmark	Roskilde Festival (Denmark); 44 samples collected over multiple days from urinals	Pooled urine collected from urinals in 3 areas; sampling conducted at different times of day and stored on ice; total of 44 urine samples collected	UPLC-HR-TOF-MS and immunoassays	77 drugs and metabolites identified: MDMA, MDA, cocaine, amphetamine, THC-COOH, ketamine (detected in 17 samples using UPLC-HR-TOF-MS but NOT detected in any urine drug-screening immunoassays); no NPS confirmed by mass spectrometry	Classical recreational drugs detected in high frequency, especially MDMA and cocaine; no NPS detected despite expectations; significant discrepancy between immunoassay and UPLC-HR-TOF-MS results	First festival study in Denmark using pooled urine analysis; demonstrated limitations of immunoassays for pooled urine screening; highlighted advantages of HRMS
([30]), Taiwan	Spring Scream festival, pop music event in Taiwan(~600,000 attendees); samples collected before, during, and after the 2011 festival week	Grab samples of wastewater influents and effluents collected daily for one week from 2 WWTPs; river water also sampled across 28 locations during festival and non-festival times	SPE + LC-MS/MS; separate methods for pharmaceuticals and illicit drugs	Detected up to 30 emerging contaminants including ketamine, MDMA, caffeine, acetaminophen, diclofenac, gemfibrozil, codeine	Substantial increases in illicit drug and pharmaceutical concentrations in wastewater during the festival, especially ketamine (138,000 ng/L) and MDMA (1267 ng/L)	River sample approach complemented wastewater data
([35]), UK and Belgium	Study analysed pooled urine from chemical toilets collected at festivals in the UK and Belgium, as well as a UK city centre; sampling locations were selected as drug use ‘hotspots’	23 grab samples taken from chemical toilets (male urinals) at UK and Belgian festivals and UK city centre locations	LC–QTOF-MS; screening performed using a comprehensive in-house HRMS spectral library with >600 entries	In total, 53 substances were detected in UK samples and 28 in Belgian samples, including NPS (e.g., methylone, mephedrone, ketamine, 4-FA, α-PVP), and classical drugs (e.g., MDMA, cocaine, amphetamine, MDA, THC-COOH)	Higher diversity of NPS in UK samples; results reflect geographical differences in recreational drug use patterns	First comparative study using pooled urine from festivals and city centres to monitor NPS prevalence; results support pooled urine as a viable tool for NPS monitoring at public events
([36]), Australia	Australian multiday summer festival in two consecutive years 2010 and 2011 (~16,700 daily attendees in 2010 and ~14,700 in 2011)	24 h composite wastewater samples collected from on-site WWTP inlet	LC-MS/MS; some targets required SPE (e.g., THC-COOH), others analysed directly after filtration	THC-COOH, MDMA, methamphetamine, cocaine, amphetamine, BZE; also detected: benzyl piperazine, mephedrone, methylone (emerging psychostimulants)	Drug use increased over festival days, peaking on final day; MDMA use significantly elevated compared to nearby urban area; methamphetamine and BZP declined from 2010 to 2011	One of the first studies to measure NPS via wastewater at festivals
([38]), Hungary	Large annual music festival held on Lake Balaton shore, Hungary (attendance: 154,000–172,000); water samples collected from multiple sites before, during, and after the festival	Grab sampling at nearshore and remote sites; samples collected April, June, July (1 day after event), August, and November across 3 years (2017, 2018, 2019)	SPE + SFC-MS/MS	Cocaine, BZE, MDMA, ketamine, methamphetamine, amphetamine, MDA, MDAI; cocaine, BZE, MDMA, and ketamine consistently detected	Illicit drug concentrations peaked 1 day post-festival and declined within 1 month; cocaine and MDMA were consistently observed across years in the highest concentrations	Weather (wind/rain) affected detection patterns; findings suggest festivals may transiently contaminate lake ecosystems with biologically active substances
([40]), Slovakia	Two Slovak music festivals in 2013: Pohoda (multicultural, Trenčín, ~30,000 attendees), Lodenica (folk/country, Piešťany, ~10,000 attendees); samples collected during and one week after each festival	24 h composite sampling at 8 WWTPs using automatic samplers; 15 min intervals; control samples taken one week later	SPE + LC-MS/MS	Methamphetamine, amphetamine, MDMA, cocaine, BZE, THC-COOH. Significant festival-associated increases: MDMA at Pohoda (up to 29 mg/day/1000 inh.); cocaine at both festivals; no THC-COOH increase	High methamphetamine use across Slovakia; cocaine and MDMA use spiked during festivals, particularly in Trenčín; drug load patterns varied by drug type and event demographics	MDMA consumption varied with music style; cannabis use widespread but not festival-associated
([39]), Czech Republic and Slovakia	Seven large-scale music festivals in Slovakia and Czech Republic during summer 2017 (metal, rock, pop, electronic, country, folk, ethnic, dance, multi-genre); total estimated attendance: >130,000	24 h composite influent wastewater samples collected from six WWTPs serving festival locations; parallel sampling at a control site with no festival activity	In-line SPE–LC-HRMS (Orbitrap)	MDMA, amphetamine, methamphetamine, cocaine, THC-COOH and 5 psychoactive pharmaceuticals (tramadol, citalopram, venlafaxine, codeine, oxazepam); significant increases in MDMA, amphetamine, and cocaine loads during festivals	Drug residues significantly elevated during festivals compared to background values and control city; MDMA showed the highest proportional increase; substance patterns varied by festival type and attendee demographics	First comparative WBE study across multiple Central European festivals; findings highlight music genre-specific drug consumption trends
([41]), Australia	Sampling conducted at a multi-day music festival in Queensland, Australia; wastewater collected daily from 25 December 2022 to 1 January 2023	Time-based composite influent wastewater samples collected every 15 min over 24 h periods using autosampler	SPE + LC-MS/MS; suspect screening for 71 substances, including NPSs	Multiple NPS identified across samples (from the total of 71 screened compounds, 7 were detected)	Seven substances were positively identified in festival wastewater	Demonstrated feasibility of NPS surveillance using WBE at temporary events; highlights challenges of interpreting unlabelled compounds using suspect screening
([53]), Australia	Five-day music festival in Queensland, Australia, studied in both 2021 and 2022; sampling covered 5 days: 2 pre-festival, 3 during-festival	Wastewater influent samples collected daily during the five-day festivals in 2021 and 2022 (samples collected using a portable refrigerated autosampler)	SPE + LC-MS/MS	MDMA, THC-COOH, cocaine, ketamine, LSD, MDA Other substances found, but not reported in surveys: MDEA, mephedrone, methylone, 3-MMC, α -D2PV, etizolam, eutylone, and N,N-dimethylpentylone	Triangulation revealed under-reporting in survey data; numerous chemicals identified in wastewater but not disclosed in surveys presumably indicate substitutions or adulterants	Emphasised strength of mixed-methods for estimating population-level drug use
([56]), Croatia	Electronic music festival in Split, Croatia; wastewater samples collected during three periods: festival week (13–19 July), peak-tourist reference week (24–30 August), and off-tourist reference week (9–15 November)	24 h time-proportional composite samples collected every 15 min from two main wastewater collectors (Stupe and Katalinica Brig)	SPE + LC-MS/MS	MDMA, cocaine, amphetamine, cannabis, heroin, morphine, codeine, tramadol, NPSs, methamphetamine	MDMA loads increased 30-fold during the festival compared to off-season and 15-fold versus peak-tourist week; smaller increases observed for AMP and cocaine (1.7-fold); NPS and MAMP detected at low levels	Highlights clear distinctions between festival and non-festival periods
([58]), Croatia	Ultra Europe electronic music festival in Split, Croatia; samples collected during 2016–2018 festivals (30 pooled urine samples were collected from portable chemical toilets); over 150,000 attendees from more than 150 countries	Syringe collection from toilets; toilets were sampled across different days and locations each year	SPE + GC/MS	46 substances detected in total: 26 classic drugs (e.g., MDMA, cocaine, amphetamine, ketamine) and 20 NPS (e.g., methcathinone, methylone, phenethylamines, cathinones, tryptamines)	Highest number of substances found on the first day of the festival; classic drug use remained stable across years, while NPS prevalence declined sharply by 2018; methcathinone was detected for the first time in 2018	Demonstrates the effectiveness of pooled urine sampling at festivals; 2018 results align with reported EU-wide decline in NPS availability (in line with EMCDDA findings)

Abbreviations: 6MAM, 6-monoacetylmorphine (metabolite of heroin consumption); AMPH, Amphetamine; BZE, Benzoylecgonine (metabolite of cocaine consumption); BZP, Benzylpiperazine; COE, Biomarker of alcohol and cocaine co-consumption; 2-ethylidene-1,5-dimethyl-3,3-diphenylpyrrolidine (primary metabolite of methadone), EDDP; EMCDDA, European Monitoring Centre for Drugs and Drug Addiction; EtS, Ethyl sulfate; FMA, 4-fluoromethamphetamine; GC-FID, Gas Chromatography with a Flame Ionisation Detector; GC/MS, Gas Chromatography-Mass Spectrometry; HPLC-MS/MS, High-Performance Liquid Chromatography–Tandem Mass Spectrometry; LC-HRMS, Liquid Chromatography–High-Resolution Mass Spectrometry; LC-MS/MS, Liquid Chromatography–Tandem Mass Spectrometry; LC-QTOF-MS, Liquid Chromatography coupled to Quadrupole Time-Of-Flight Mass Spectrometry; Limit of Quantification, LOQ; LSD, Lysergic Acid Diethylamide; MAMPH, Methamphetamine; MDA, 3,4-methylenedioxyamphetamine; MDAI, 5,6-methylenedioxy-2-aminoindane; MDMA, 3,4-Methylenedioxymethamphetamine (commonly known as ecstasy or molly); MDPV, 3,4-methylenedioxypyrovalerone; MMC, 3/4-methylmethcathinone; NPS, New Psychoactive Substances; QTOF-MS, Quadrupole Time-Of-Flight Mass Spectrometer; SFC-MS/MS, Supercritical Fluid Chromatography-Tandem Mass Spectrometry; SPE, Solid-Phase Extraction; SPE-LC-MS/MS, Solid-Phase Extraction Liquid Chromatography–Tandem Mass Spectrometry; SPE–UHPLC–MS/MS, Solid-Phase Extraction Ultra High-Performance Liquid Chromatography–Tandem Mass Spectrometry; STP, Sewage Treatment Plants; THC-COOH, 11-nor-9-carboxy-tetrahydrocannabinol which is the inactive metabolite of the psychoactive substance Δ^9^-tetrahydrocannabinol (THC) that is contained in cannabis; UPLC-HR-TOF-MS, Ultra-Performance Liquid Chromatography High-Resolution Time-Of-Flight Mass Spectrometry; WBE, Wastewater-Based Epidemiology; WWLS, Wastewater Lift Station; WWTP, Wastewater Treatment Plants; α-D2PV, Novel synthetic cathinone, part of the broader class of NPS.

**Table 2 behavsci-15-01672-t002:** Detailed breakdown of analytical techniques.

Analytical Technique	Explanation	Application in This Review
LC-MS/MS (Liquid Chromatography–Tandem Mass Spectrometry)	A highly sensitive analytical method that separates substances in a liquid sample and identifies them based on their mass-to-charge ratio. It is widely used for detecting and quantifying trace levels of drugs in complex biological matrices such as wastewater.	Used in the majority of studies (16 out of 23) to detect commonly used substances such as MDMA, cocaine, and amphetamines at very low concentrations. Primary quantitative method across studies including [7] ([7]), [8] ([8]), [10] ([10]), [12] ([12]), and others.
LC-HRMS (Liquid Chromatography–High-Resolution Mass Spectrometry)	An advanced analytical technique that provides highly accurate mass measurements and superior compound identification capabilities compared to conventional MS. Essential for NPS detection and confirmation.	Used in 5 out of 23 studies for NPS identification including [5] ([5]), [9] ([9]), [11] ([11]), [22] ([22]), and [39] ([39]).
SPE (Solid-Phase Extraction)	A preparatory technique used to isolate and concentrate analytes of interest from a sample by passing it through a solid absorbent material. This process reduces interference and improves reliability.	Standard sample preparation in 17 out of 23 studies to purify wastewater before further analysis using LC-MS/MS or related methods. Essential for achieving detection limits required for trace-level festival drug monitoring.
GC-MS (Gas Chromatography–Mass Spectrometry)	An analytical technique that vaporises chemical substances and separates them by their physical and chemical properties before identifying them through mass spectrometry.	Applied in 3 studies ([58]; [21]; [38] using SFC-MS/MS variant) to identify classic drugs of abuse, particularly those that are volatile or stable at high temperatures. Less common but effective for specific compound classes.
QTOF-MS (Quadrupole Time-of-Flight Mass Spectrometry)	A hybrid mass spectrometry technique combining quadrupole ion selection with time-of-flight mass analysis, providing high mass accuracy and the ability to perform both targeted and untargeted analysis.	Used in 2 studies ([21]; [35]) for comprehensive screening of classical drugs and NPS.
UPLC-HR-TOF-MS (Ultra-Performance Liquid Chromatography High-Resolution Time-Of-Flight Mass Spectrometry)	The most advanced form of mass spectrometry that allows for comprehensive screening of both known and emerging substances.	Specialised technique employed in [26] ([26]) focusing on detection of emerging drug trends and novel substances. Demonstrated clear advantages over traditional immunoassays for pooled urine analysis, but showed limitations for NPS detection despite expectations.

Abbreviations: MDMA, 3,4-Methylenedioxyamphetamine (commonly known as ecstasy or molly); NPS, New Psychoactive Substances; SFC-MS/MS, Supercritical Fluid Chromatography–Tandem Mass Spectrometry.

**Table 3 behavsci-15-01672-t003:** Detailed breakdown of regional patterns in festival drug consumption.

Country	Preferred Drugs	Studies Referenced	Key Regional Findings
Australia	Traditional Drugs: MDMA (festival-dominant), MDA, cannabis, methamphetamine, cocaine, amphetamine, ketamine NPS: α-D2PV, N-ethylpentylone, mephedrone, methylone Pattern: MDMA consumption “substantially higher” at festivals vs. residential areas	[5] ([5]); [13] ([13]); [36] ([36]); [41] ([41]); [53] ([53])	Festival vs. Residential Differences: Cannabis > MDMA > Methamphetamine > Cocaine (festival) vs. Cannabis > Methamphetamine > MDMA > Cocaine (urban). First country to apply DOE optimisation for 71 NPS detection.
Belgium	Traditional Drugs: MDMA (highest concentrations), MDA, cocaine, amphetamine, cannabis, ketamine NPS: 4-FMC, 4-MEC, α-PVP, 2C-B, 4-FA	[9] ([9]); [21] ([21]); [35] ([35])	Lower NPS diversity compared to UK. Critical safety finding: Dangerous cardiac medications used as cocaine adulterants (flecainide and amloidipine) MDMA consistently high relative to other traditional drugs.
Brazil	Traditional Drugs: Cocaine (substantially higher than MDMA), MDMA, BZE, cocaethylene Pattern: Festival-associated increases	[23] ([23])	Cocaine-dominant pattern (unusual compared to European MDMA dominance). Wastewater data revealed higher usage levels than national averages.
Croatia	Traditional Drugs: MDMA, cocaine, amphetamine, cannabis NPS: phenethylamines and cathinones (primarily)	[58] ([58]); [56] ([56])	[58] ([58]): 46 total substances (26 classic + 20 NPS) from pooled urine 2016–2018. [56] ([56]): Comprehensive wastewater analysis with 30-fold MDMA increase, stable cannabis/heroin/opioids/nicotine.
Czech Republic	Traditional Drugs: Methamphetamine-dominant (historical legacy), cannabis, MDMA, cocaine, amphetamine Psychoactive pharmaceuticals: tramadol, citalopram, venlafaxine, codeine, oxazepam	[39] ([39])	Key pattern: Shared post-Communist methamphetamine dominance with Slovakia Festival findings: Similar drug consumption patterns to Slovakia during music festivals
Denmark	Traditional Drugs: MDMA, MDA, cocaine, amphetamine, cannabis, ketamine NPS: Lower than expected based on seizure data	[26] ([26])	Ketamine detected in 17 samples by HRMS but zero detection by immunoassays (demonstrated analytical method superiority).
France	Traditional Drugs: Cocaine, BZE, MDMA, cannabis, cocaethylene	[15] ([15])	No specific festival effect observed (attributed to event’s public nature and diverse audience).
Hungary	Traditional Drugs: MDMA, cocaine, BZE, ketamine, amphetamine, methamphetamine, MDA, MDAI	[38] ([38])	Environmental contamination of lake ecosystem with psychoactive substances. Weather effects on detection patterns observed.
Norway	Traditional Drugs: MDMA (lower than cocaine), cannabis, cocaine, amphetamine/methamphetamine sufficient for quantification (unique pattern), NPS: Various substances detected in pooled urine	[9] ([9]); [22] ([22])	Different amphetamine-type pattern: higher amphetamine/methamphetamine usage in Norway (high enough to quantify accurately), while other countries (UK, Belgium) had levels too low for reliable quantification ([9]). Contradictory findings: “neither amphetamine, methamphetamine, ketamine, cathinones, phenethylamines nor other NPS were used by a significant proportion of the participants” ([22])
Portugal	Traditional Drugs: Cannabis (10x increase during festival), MDMA, cocaine, amphetamine NPS: 3,4-DMMC, α-methyltryptamine, methcathinone, mephedrone, buphedrone, ketamine	[9] ([9])	Festival-specific NPS use: Zero NPS in non-festival samples. Cannabis showed most dramatic increase (10-fold) during festival period.
Serbia	Traditional Drugs: MDMA clearly most prevalent during festivals, amphetamine, cocaine, cannabis Pattern: MDMA-dominant festival culture	[9] ([9])	MDMA-centric festival culture with clear dominance over other substances during electronic music events. 30-fold MDMA increase during festivals.
Slovakia	Traditional Drugs: Methamphetamine-dominant (historical legacy), cannabis, MDMA, cocaine, BZE NPS: cathinones and phenethylamines (primarily) Psychoactive pharmaceuticals: tramadol, citalopram, venlafaxine, codeine, oxazepam Geographic Pattern: Highest cocaine in Bratislava, highest meth in rural areas	[11] ([11]); [12] ([12]); [40] ([40]); [39] ([39])	Post-Communist legacy: “EMCDDA counts Slovakia and Czech Republic among countries with dominant methamphetamine consumption” ([39]) due to historical availability. Festival variety: Different substances for different music genres.
Spain	Traditional Drugs: Cocaine, MDMA, amphetamine, cannabis, BZE, MDA Temporal: Festival-associated increases	[10] ([10]) [9] ([9])	Significant MDMA increases during festival periods. Wastewater treatment efficiency challenged during peak use.
Switzerland	Traditional Drugs: MDMA, amphetamine, methamphetamine, cocaine, cannabis, ketamine, BE, methadone, codeine NPS: Mephedrone and methylone (detected, but not quantified)	[7] ([7]) [8] ([8])	Festival monitoring showed dramatic increases. Per capita increases: MDMA and cocaine increased substantially after events
Taiwan	Traditional Drugs: Mainly ketamine and MDMA Pattern: Detected 30 emerging substances with concentrations peaking during festival	[30] ([30])	Environmental focus: Spring Scream Festival monitoring of 30 sampling sites (28 river samples + 2 WWTPs). Pollution control perspective rather than epidemiological focus.
UK	Traditional Drugs: MDMA, cocaine, amphetamine, cannabis, ketamine NPS: Highest NPS diversity globally (53 substances detected) Temporal: More NPS in 2015 vs. 2016; ketamine concentrations increased 2016	[9] ([9]); [35] ([35])	Global NPS leader: Highest diversity of novel substances. Geographic advantage: Comprehensive monitoring of both festivals and urban “drug hotspots”.

Abbreviations: BZE, Benzoylecgonine (metabolite of cocaine consumption); EMCDDA, European Monitoring Centre for Drugs and Drug Addiction; HRMS, High-Resolution Mass Spectrometry; MDA, 3,4-methylenedioxyamphetamine; MDAI, 5,6-methylenedioxy-2-aminoindane; MDMA, 3,4-Methylenedioxyamphetamine (commonly known as ecstasy or molly); NPS, New Psychoactive Substances; WWTP, Wastewater Treatment Plants; α-D2PV, Novel synthetic cathinone, part of the broader class of NPS.

**Table 4 behavsci-15-01672-t004:** Synopsis of the main findings.

Theme	Main Findings
Sampling methodologies and analytical approaches	Two primary approaches: wastewater sampling and pooled urine sampling with LC-MS/MS and HRMS analytical methods Pooled urine sampling offers higher analyte concentrations but limited demographic representation
Substance detection patters	MDMA dominance across diverse geographical contexts and festival types Profound regional variations Music genre influences Substantially higher consumption during festival periods compared to baseline
Public health and harm reduction applications	Population-level anonymous monitoring enables objective evidence-based intervention design Early warning system potential for NPS and dangerous adulterants Temporal limitations: Lag between sample collection and result availability limits real-time harm reduction responses

Abbreviations: HRMS, High-Resolution Mass Spectrometry; LC-MS/MS, Liquid Chromatography–Tandem Mass Spectrometry; MDMA, 3,4-Methylenedioxyamphetamine (commonly known as ecstasy or molly); NPS, New Psychoactive Substances.

## Data Availability

Not applicable.

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
