# Peer review of "Wastewater Analyses for Psychoactive Substances at Music Festivals: A Systematic Review"

_behavsci, 2025, doi:10.3390/bs15121672_

Round 1
Reviewer 1 Report
Comments and Suggestions for Authors
I would like to thank the editor for inviting me to review this manuscript. I appreciate the opportunity to provide feedback on this topic, and I hope my comments will be helpful in strengthening the clarity, rigor, and public health relevance of the review. While the authors demonstrated that they conducted a study with a strong methodology, I identified some important concerns that need to be addressed prior to publication.
Throughout the abstract and the introduction, I found that the following were unclear or unjustified:
- Rationale for the review:
- The justification for conducting a systematic review on this topic is not well articulated. While the authors note that monitoring drug use is challenging, it is unclear how synthesizing the literature will concretely improve field surveillance or public health interventions.
- Focus on wastewater and urine only:
- The review is limited to studies using wastewater or pooled urine, which may not capture the full spectrum of substance use. Drugs can be contaminated, and individuals may believe they are consuming different substances than those detected. Including qualitative or mixed-method studies could provide valuable contextual insights.
- Association with music genres:
- Linking substance use to music genres is potentially stigmatizing. The practical utility of such associations for public health interventions is unclear.
Introduction
- Definition of substances:
- The statement “substances devoid of any medicinal purpose, synthesized and consumed mainly for their psychoactive effects” is inaccurate. Most psychoactive substances are used for self-regulation (emotional, psychological, social), and labeling them as “without medicinal purpose” can stigmatize users.
- The terminology “illicit drugs” or “drugs of abuse” , and the overall discussion about substance use disorder and mental health is questionable given that the study is about wastewater analyses. - The definition provided here is somewhat outdated and potentially stigmatizing. Referring to all substances that are illegal or “manufactured for misuse” as “illicit drugs” or “drugs of abuse” implies that their use is inherently problematic or devoid of potential benefits. Contemporary perspectives in addiction research increasingly recognize that substance use exists along a continuum and that legality does not determine the nature or outcomes of use. I would encourage the authors to adopt a more neutral and precise terminology
- Links between to wastewater research and substance use disorder /mental health:
- The link between substance use disorder and the rationale for wastewater studies is unclear. Wastewater analysis does not provide individual-level data on misuse or mental health outcomes, so the transition should be clarified and less stigmatizing.
- Statements suggesting that substances detected in wastewater are “misused” are misleading. Presence in wastewater does not indicate misuse; legality alone does not define problematic use (e.g., alcohol is legal yet highly harmful).
- Claims that wastewater analysis can improve diagnosis or management of drug-induced mental health disorders overstate the utility of these methods.
- Music genre associations:
- Linking substance use to musical genres risks reinforcing stereotypes. Evidence suggests that social contexts, not music per se, shape use patterns. The authors should consider recent research that nuances these associations.
- Law enforcement applications:
- Framing WBE as a tool for policing risks conflicting with public health principles. Harm reduction should remain the central focus; emphasizing detection and control could inadvertently reinforce punitive approaches.
Materials and Methods
Methodology is rigorous and clearly presented. No major critiques.
Results and Data Presentation
- Music genre bias:
- Many studies analysed did not identify a specific music genre or included mixed genres, yet the review sometimes links substances to genres. This may introduce bias, reinforcing misleading associations.
- WBE cannot differentiate between recreational or problematic use; language should avoid implying misuse or risk profiles.
Limitations
- Music genre comparisons:
- Suggesting future research focus on genre-specific differences is questionable. Such comparisons risk stigmatizing attendees without clear public health benefit. Future studies should prioritize broader patterns of use and contextual factors rather than contrasting genres.
- The reasons for conducting this systematic review remain unclear: is the goal harm reduction, surveillance, or policing? Implications for practice are underdeveloped.
- The implications section largely summarizes research design points rather than providing actionable recommendations for harm reduction, festival interventions, or public health surveillance.
Author Response
Throughout the abstract and the introduction, I found that the following were unclear or unjustified:
Rationale for the review:
The justification for conducting a systematic review on this topic is not well articulated. While the authors note that monitoring drug use is challenging, it is unclear how synthesizing the literature will concretely improve field surveillance or public health interventions.
- We thank the reviewer for this comment. We have toned down the wording of the last sentence of section 1.4. In the amended manuscript, we are now writing: “Wastewater analysis can contribute valuable data for public health surveillance, and this information might hopefully support the development of more effective, evidence-based policies and improve preparations for medical and psychological support for musical events with harm reduction as the central focus.” We would like to emphasise that we are talking about “wastewater analysis” in general, not about our systematic review in this sentence.
Additionally, we changed the wording of the aim of this review in section. Section 1.5 now reads: “Therefore, the aim of this systematic review is to critically evaluate and synthesize the existing literature on wastewater analysis to measure recreational drug consumption at mu-sic festivals, providing an in-depth examination in this specific context of festivals where music is the main feature.”
We hope that these more modest formulations clarify that this review is not aiming to specifically or directly improve field surveillance or public health interventions. It just summarises the published literature.
Focus on wastewater and urine only:
The review is limited to studies using wastewater or pooled urine, which may not capture the full spectrum of substance use. Drugs can be contaminated, and individuals may believe they are consuming different substances than those detected. Including qualitative or mixed-method studies could provide valuable contextual insights.
- We agree with the reviewer. Therefore, we included the following sentence in the discussion in section 4.2.1: “We restricted the review to wastewater analysis. Therefore, we did not capture many important aspects of substance use at music festivals. Wastewater analysis measures the consumed substances of the group or a subgroup of festival goers, not the drug consump-tion of one individual. Moreover, as drugs can be contaminated, individuals might believe they are consuming different substances than those detected. To investigate such dis-crepancies, mixed-method studies including individual qualitative and quantitative in-terviews as well as individual drug tests could provide valuable contextual insights.”
Association with music genres:
Linking substance use to music genres is potentially stigmatizing. The practical utility of such associations for public health interventions is unclear.
- To avoid unjustified conclusions about the entirety of festival goers, we have inserted the following sentence into section 4.2.6.: “Wastewater analysis uses pooled specimens and can only provide a positive test of a drug or the concentration of a drug within the whole sample, not for the individual per-son. This method is unsuitable to ascertain the proportion of festival goers which consume a substance. A small subgroup of festival visitors or even a single visitor can influence the overall result. Thus, data derived from wastewater analysis at music festivals as well as secondary research studies based on such analyses must take this methodological shortcoming into account to avoid conclusions about individual festival participants or the totality of festival goers.”
Introduction
Definition of substances:
The statement “substances devoid of any medicinal purpose, synthesized and consumed mainly for their psychoactive effects” is inaccurate. Most psychoactive substances are used for self-regulation (emotional, psychological, social) and labelling them as “without medicinal purpose” can stigmatize users.
- We have deleted this sentence.
The terminology “illicit drugs” or “drugs of abuse”, and the overall discussion about substance use disorder and mental health is questionable given that the study is about wastewater analyses. - The definition provided here is somewhat outdated and potentially stigmatizing. Referring to all substances that are illegal or “manufactured for misuse” as “illicit drugs” or “drugs of abuse” implies that their use is inherently problematic or devoid of potential benefits. Contemporary perspectives in addiction research increasingly recognize that substance use exists along a continuum and that legality does not determine the nature or outcomes of use. I would encourage the authors to adopt a more neutral and precise terminology
- We inserted the following sentence in section 1.1.: “). However, these terms focus on the current legal status of the drug, but not its effects or other relevant aspects. Therefore, we will refer to non-prescribed drugs as “recreational drugs.“ We also changed the term “illicit” throughout the manuscript except from titles of cited articles and the search terms.
Links between to wastewater research and substance use disorder /mental health:
The link between substance use disorder and the rationale for wastewater studies is unclear. Wastewater analysis does not provide individual-level data on misuse or mental health outcomes, so the transition should be clarified and less stigmatizing.
- We deleted the sentence: “This understanding may contribute to improved diagnosis and management of drug-induced mental health disorders.”
Statements suggesting that substances detected in wastewater are “misused” are misleading. Presence in wastewater does not indicate misuse; legality alone does not define problematic use (e.g., alcohol is legal yet highly harmful).
- We deleted the sentence “Substance misuse transpires when individuals utilize drugs for reasons divergent from their intended purposes,” and we have reformulated all sentences which included the word “misuse” or “misused”.
Claims that wastewater analysis can improve diagnosis or management of drug-induced mental health disorders overstate the utility of these methods.
- We deleted the sentence: “This understanding may contribute to improved diagnosis and management of drug-induced mental health disorders.”
Music genre associations:
Linking substance use to musical genres risks reinforcing stereotypes. Evidence suggests that social contexts, not music per se, shape use patterns. The authors should consider recent research that nuances these associations.
- We inserted the sentence: “WBE might detect specific drug use patterns associated with the musical genre, the cultural milieu and the geographic region (Ort et al., 2014; Wright et al., 2020; Hoegberg et al., 2018; Bijlsma et al. 2020). However, regarding the causality, the association between the consumption of a specific drug with a musical genre might be an epiphenomenon that is driven by the social context, not the type of music per se.”
We are also writing in section 4.1.2.: “As only some of the analyzed studies compared specific music genres or included mixed genres, definitive conclusions regarding the association between music genres and consumed drugs should not be made. Preliminary conclusions may introduce bias, reinforcing misleading associations” and in section 4.3.: “Sixth, future research on genre-specific differences risks stigmatizing attendees without clear public health benefit. Future studies should therefore prioritize broader patterns of use and contextual factors rather than simply contrasting genres. In this respect, we would like to mention that drug and alcohol consumption are commonplace at various types of festivals, not only those with a focus on music. Bade et al. (2020), for example, collected wastewater over several consecutive weeks from a location where a school-leaver festival was held and found significant changes in MDMA and MDA concentrations over the festival week.”
Law enforcement applications:
Framing WBE as a tool for policing risks conflicting with public health principles. Harm reduction should remain the central focus; emphasizing detection and control could inadvertently reinforce punitive approaches.
- We deleted the paragraph about law enforcement. Instead, we are now writing in section 1.4.: “WBE might detect specific drug use patterns associated with the musical genre, the cultural milieu and the geographic region (Ort et al., 2014; Wright et al., 2020; Hoegberg et al., 2018; Bijlsma et al. 2020). However, regarding the causality, the association between the consumption of a specific drug with a musical genre might be an epiphenomenon that is driven by the social context, not the type of music per se.
- Wastewater analysis can contribute valuable data for public health surveillance, and this information might hopefully support the development of more effective, evi-dence-based policies and improve preparations for medical and psychological support for musical events with harm reduction as the central focus.”
Materials and Methods
Methodology is rigorous and clearly presented. No major critiques.
- We thank the reviewer for the positive feedback.
Results and Data Presentation
Music genre bias:
Many studies analysed did not identify a specific music genre or included mixed genres, yet the review sometimes links substances to genres. This may introduce bias, reinforcing misleading associations.
- In the paragraph regarding music genres, we have cited the specific studies that included a specific genre or compared specific genres. Therefore, we do not fully agree with the reviewer. However, we had already written in section 4.1.2.: “However, the observation by Lai et al. (2013) that “MDMA use is not limited to the electronic/techno dance scenes” indicates that while genre associations exist, they may not be deterministic, with festival culture and attendee expectations potentially mediating these relationships.” Furthermore, we have added the following sentence into section 4.1.2. of the revised manuscript: “As only some of the analyzed studies compared specific music genres or included mixed genres, definitive conclusions regarding the association between music genres and consumed drugs should not be made. Preliminary conclusions may introduce bias, reinforcing misleading associations.”
WBE cannot differentiate between recreational or problematic use; language should avoid implying misuse or risk profiles.
- We have deleted the term “misuse” throughout the manuscript.
Limitations
Music genre comparisons:
Suggesting future research focus on genre-specific differences is questionable. Such comparisons risk stigmatizing attendees without clear public health benefit. Future studies should prioritize broader patterns of use and contextual factors rather than contrasting genres.
- In addition to what we have already amended in this regard in response to previous comments on the topic, we have inserted the following paragraph into section 4.3. on future research: “Sixth, future research on genre-specific differences risks stigmatizing attendees without clear public health benefit. Future studies should therefore prioritize broader patterns of use and contextual factors rather than simply contrasting genres. In this respect, we would like to mention that drug and alcohol consumption are commonplace at various types of festivals, not only those with a focus on music. Bade et al. (2020), for example, collected wastewater over several consecutive weeks from a location where a school-leaver festival was held and found significant changes in MDMA and MDA concentrations over the festival week.”
The reasons for conducting this systematic review remain unclear: is the goal harm reduction, surveillance, or policing? Implications for practice are underdeveloped.
- We have deleted all sentences relating to policing and law enforcement completely. We believe that we have made it clearer that “… the aim of this systematic review is to critically evaluate and synthesize the existing literature on wastewater analysis to measure recreational drug consumption at music festivals, providing an in-depth examination in this specific context of festivals where music is the main feature.” We have written this in section 1.5. “Aim”.
We also deleted the paragraph “… real-time monitoring capabilities should be developed to enable immediate public health responses during festival events. Advanced analytical technologies and automated alert systems could transform festival monitoring from retrospective analysis into proactive harm reduction, potentially preventing overdoses and dangerous substance combinations through early warning systems” in the future directions section.
The implications section largely summarizes research design points rather than providing actionable recommendations for harm reduction, festival interventions, or public health surveillance.
- We disagree with the reviewer, because the whole paragraph 4.1.3. ”Public Health and Harm Reduction Applications” addresses this point. We would like to make them particularly aware of our statement that “… successful implementation depends on addressing current temporal, technological, and coordination limitations through sustained methodological development and stakeholder engagement.” However, we have added the following sentence at the end of section 4.1.3.: “Stakeholders from all involved groups (e.g., festival goers, festival organizers, musicians, political and cultural representatives, physical and mental health experts, WBE experts and environmental associations) should discuss and develop actionable recommendations for public health surveillance, harm reduction and festival interventions.”
Reviewer 2 Report
Comments and Suggestions for Authors
I have reviewed the manuscript entitled, ‘Wastewater Analyses for Psychoactive Substances at Music Festivals: A Systematic Review,’ according to the journal’s criteria. Overall, the manuscript is well-written, the method is clearly described and most of the conclusions, limitations and future directions are justified. The main issue I would like the authors to consider is the choice of databases included in the literature search. SCOPUS seems to be an obvious omission extending to the environmental sciences and may be the reason at least 2 studies that I am aware of have been missed (Aberg et al, 2022, Env Res, The environmental release and ecosystem risks of illicit drugs during Glastonbury Festival. Bade et al, 2020, Drug and Alc Dep, What is the drug of choice of young festivalgoers?). There may be several others, so without a more comprehensive search, this manuscript cannot claim to have exhaustively extracted relevant literature.
Below I include comments relating to specific parts of the manuscript. This is a long list of comments, but it is a review article and I hope the authors find the comments useful. Line numbers refer to those in the PDF available for review.
Line 29. From the main body of the manuscript, it is clear that the authors imply NPS metabolites have limited availability. NPS reference materials are generally available for forensic use. Please rephrase the statement.
Line 37-56. This text seems overly expansive for the likely users of this review. Please condense.
Line 79-80. Please rephrase. WBE is not a method.
Line 94. I do not think an 11-year-old citation is appropriate for the current state of knowledge on NPS metabolism and stability.
Line 95. The authors need to be careful with statements about benefits in this context. Portaloos collect more than just urine samples and always contain sanitiser and deodoriser. The impact of these chemicals on drug residue stability has not been demonstrated, to my knowledge.
Line 100. I doubt local festivals are any less sites of above-average drug use. Why the emphasis on international?
Line 123-124. While the statement is reasonable, it is also true that polydrug use is a phenomenon related to harm. WBE is unable to show this behaviour. Festival testing is mostly done after the event (despite claims to the contrary), which means the harm is already done. This is definitely true for drug measured in wastewater. Thus, the statement should rather be framed around formulating policy and interventions, instead of benefit to the festival attendees as implied.
Line 138. As above. Wastewater testing is unlikely to reduce harm during an event.
Line 157. As suggested under general remarks, SCOPUS is an omission. WBE findings are rarely reported in clinical journals.
Line 229 and 234, Fig 1. Screening – Reports excluded (n=3) Full text not available. Please clarify. Were these requested from the corresponding author?
Line 246, Table 1. Bade et al. Was alpha D2PV the only drug reported in this study?
Table 1. Please check and correct use of BE as abbreviation of benzoyl ecgonine. In some parts, BZE is used (e.g. Bijlsma 2014). It may also be better to be consistent when THC COOH or cannabis is used. Presumably the methods all targeted the metabolite, not the herb.
Line 284. Please check the duration. This seems an unlikely high frequency.
Line 291. I do not follow the logic. A grab sample by implication represents a single time point. How does the collector know if it represents peak use or not? Drugs are not excreted immediately after use.
Line 301. Please clarify if the samples were collected during the event and analysed on site. This is not my understanding from the original text. Thus, real-time is questionable.
Table 2 (bottom of page). QToF and HR-ToF are the same thing, just different terms.
Line 323. The approach is also the most selective for targeted analysis.
Line 329. Credit to Bade et al, but this is not very respectful towards many other labs that have done earlier work on cell assays to identify drug targets in wastewater (just not festivals - but does that matter?).
Line 333. It is worth mentioning that the technology used for this study (Nadarajan) is not widely available to other labs.
Table 3. Considering the topic of the Table and heading of column (Preferred substances), the authors are advised to specify the parent drug, not the metabolite to avoid misinterpretation of the results.
Line 548-549. Sampling frequency is only a consideration when sewer infrastructure is in place. In urinals, portable toilet blocks or portaloos, it is more about drug residue stability. Once excreted, the sample remains there until the unit is emptied – typically after the event or at night.
Line 551. This is a sweeping statement that is carried forward throughout the manuscript. In fact, portable toilet blocks are often designated male/female and could be sampled when such infrastructure is in place. This obviously varies from country to country and festival to festival, but the authors should not generalise.
Line 572. This is open to debate. The equipment used for direct injection is cost prohibitive for most labs and requires constant maintenance to maintain sensitivity. I challenge this approach as universally applicable or recommended. A SPE manifold cost next to nothing and can be operated by a person with minimal training. It is the mass spec that requires specialist operators and resources.
Line 685. The authors recognise the limitation of their choice of databases. I suggest adding SCOPUS.
Line 706. While this was clearly identified as a limitation in the literature search, it can be overcome with suitable infrastructure. The authors may include that as a future direction (line 814). Please note that movable toilet blocks are standard at festivals in parts of the world and not innovative at all.
Line 734-738. This point is highly relevant. The authors should also mention the use of sanitiser chemicals in portaloos.
Line 741. The authors should perhaps consider if the lack of commercially available NPS is really such an issue. Whether a NPS is excreted or disposed of at the site, it shows the drug was intended to be used or could be used by other festival attendees. Isn’t the harm messaging the same?
Line 771-779. Another point to consider is the rate of drug clearance from the body. During a festival event spanning less than a day, there is a high chance that drugs used during the festival may only be metabolised and excreted after the event, at a different site. In other words, interpreting results become complicated. It is also possible that drugs excreted at the site were consumed well prior to the event. Drug loads in wastewater from festivals should be interpreted with caution.
Line 834-838. I generally disagree with the statement about real-time monitoring, assuming the authors are alluding to wastewater analysis. Considering the analytical techniques described in this review required to measure the types of harmful drugs, real-time detection using wastewater is simply not possible. Additionally, it takes time for drugs to get metabolised and cleared from the body into the wastewater. The authors can calculate for themselves the time required for a typical LC/MS analysis and how few samples can be done in a day if sampling happens at a reasonable frequency.
Line 839. Please check the reference on the Glastonbury festival where the authors do discuss soil and water contamination.
Author Response
The main issue I would like the authors to consider is the choice of databases included in the literature search. SCOPUS seems to be an obvious omission extending to the environmental sciences and may be the reason at least 2 studies that I am aware of have been missed (Aberg et al, 2022, Env Res, The environmental release and ecosystem risks of illicit drugs during Glastonbury Festival. Bade et al, 2020, Drug and Alc Dep, What is the drug of choice of young festivalgoers?). There may be several others, so without a more comprehensive search, this manuscript cannot claim to have exhaustively extracted relevant literature.
- We thank the reviewer for this comment. Accordingly, we have updated the search including SCOPUS. This search led to the inclusion of one more article by Berset et al. 2010. The inclusion of this article led to various changes in the text and the tables throughout the manuscript. However, the two articles identified by the reviewer did not meet our inclusion criteria. Aberg et al. (2022) did not collect wastewater bus water from rivers and Bade et al. (2020) reported wastewater analysis for a school-leaver festival, not a music festival. However, these two studies are very relevant, and therefore, we inserted in section Aberg et al. (2022) in section 1.3, and mentioned Bade et al. (2020) in section 4.3.
Below I include comments relating to specific parts of the manuscript. This is a long list of comments, but it is a review article and I hope the authors find the comments useful. Line numbers refer to those in the PDF available for review.
Line 29. From the main body of the manuscript, it is clear that the authors imply NPS metabolites have limited availability. NPS reference materials are generally available for forensic use. Please rephrase the statement.
- We have deleted the statement on limited reference standards for NPS in the abstract.
Line 37-56. This text seems overly expansive for the likely users of this review. Please condense.
- We have shortened this paragraph by 50%.
Line 79-80. Please rephrase. WBE is not a method.
- We changed the wording to: “Wastewater-based epidemiology (WBE) is a population-level monitoring approach that utilizes the examination of chemical and biological indicators in wastewater to assess community-wide exposure to diverse substances …” (section 1.2).
Line 94. I do not think an 11-year-old citation is appropriate for the current state of knowledge on NPS metabolism and stability.
- We deleted the sentence: “Therefore, the examination of wastewater for tracking recreational drug consumption trends has intensified in recent years; nonetheless, this methodology encounters constraints regarding NPSs owing to inadequate understanding of their molecular structures, strength, and breakdown by bacteria in wastewater (Archer et al., 2014).”
Line 95. The authors need to be careful with statements about benefits in this context. Portaloos collect more than just urine samples and always contain sanitiser and deodoriser. The impact of these chemicals on drug residue stability has not been demonstrated, to my knowledge.
- In principle, we agree that this is an important point to raise. However, we felt that this thought fits well with the discussion. Therefore, we added the sentences: “Such protocols should consider that portaloos also collect sanitizers and deodorizers which might impact drug residue stability.” to section 4.3.
Line 100. I doubt local festivals are any less sites of above-average drug use. Why the emphasis on international?
- We changed the sentence and are now writing: “Current literature suggests that music festivals can be venues where novel NPS and other recreational drugs are consumed…” (section 1.2).
Line 123-124. While the statement is reasonable, it is also true that polydrug use is a phenomenon related to harm. WBE is unable to show this behaviour. Festival testing is mostly done after the event (despite claims to the contrary), which means the harm is already done. This is definitely true for drug measured in wastewater. Thus, the statement should rather be framed around formulating policy and interventions, instead of benefit to the festival attendees as implied.
- We amended the wording of the whole paragraph accordingly and inserted the sentence: “Currently, WBE cannot measure polydrug use which has the strongest association with harm. As festival testing is mostly done after the event, it can only inform the planning for the next, but usually not the medical and psychological support for an ongoing event” (section 4.3).
Line 138. As above. Wastewater testing is unlikely to reduce harm during an event.
- We changed this last paragraph of section 1.4. to “Wastewater analysis can contribute valuable data for public health surveillance, and this information might hopefully support the development of more effective, evi-dence-based policies and improve preparations for medical and psychological support for musical events with harm reduction as the central focus.” See also our response to reviewer 1.
Line 157. As suggested under general remarks, SCOPUS is an omission. WBE findings are rarely reported in clinical journals.
- We redid the literature review and included SCOPUS. This procedure led to the inclusion of an additional study (Berset et al. 2010). We thank the reviewer for this suggestion.
Line 229 and 234, Fig 1. Screening – Reports excluded (n=3) Full text not available. Please clarify. Were these requested from the corresponding author?
- But we did not get a response from the corresponding authors. Thus, our amended sentence reads: “Out of these, 33 articles were evaluated at the full-text level, and out of 9 excluded, 6 were discarded due to the unavailability of the full text despite our attempt to contact the corresponding authors.”
Line 246, Table 1. Bade et al. Was alpha D2PV the only drug reported in this study?
- It was a case study on D2PV and its metabolites.
Table 1. Please check and correct use of BE as abbreviation of benzoyl ecgonine. In some parts, BZE is used (e.g. Bijlsma 2014). It may also be better to be consistent when THC COOH or cannabis is used. Presumably the methods all targeted the metabolite, not the herb.
- We thank the reviewer very much for detecting this mistake. In the amended manuscript, we abbreviate Benzoylecgonine as BZE in the table, and we use “THC COOH” throughout table 1.
Line 284. Please check the duration. This seems an unlikely high frequency.
- We explain the approach more specifically in the revised version of the manuscript. We are now writing in section 3.3.1.: “Several studies took samples every 15 minutes. Puljevic et al. (2024) used a battery-powered sampling device which collected ∼1 mL subsamples every 10 s into one composite sample over the 24-h.”
Line 291. I do not follow the logic. A grab sample by implication represents a single time point. How does the collector know if it represents peak use or not? Drugs are not excreted immediately after use.
- We deleted the sentence: “While grab samples provided snapshots of drug presence at specific time points, they were recognized as potentially less representative of overall consumption patterns compared to composite approaches, though they remained valuable for detecting peak usage periods and identifying emerging substances.”
Line 301. Please clarify if the samples were collected during the event and analysed on site. This is not my understanding from the original text. Thus, real-time is questionable.
- We deleted the superfluous sentence: “This approach was particularly effective for real-time monitoring of substance use patterns and detection of NPSs due to the higher achievable concentrations.”
Table 2 (bottom of page). QToF and HR-ToF are the same thing, just different terms.
- We disagree with the reviewer. We believe that UPLC-HR-TOF-MS and QTOF-MS are not the same, because QTOF-MS refers to a specific type of mass spectrometer, while UPLC-HR-TOF-MS describes a complete analytical system that uses a high-resolution time-of-flight (HR-TOF) mass spectrometer that is preceded by ultra-performance liquid chromatography (UPLC) which is a separation method that prepares the sample, while the mass spectrometer (like a Q-TOF or a stand-alone TOF) is the detection method.
Line 323. The approach is also the most selective for targeted analysis.
- We added this information.
Line 329. Credit to Bade et al, but this is not very respectful towards many other labs that have done earlier work on cell assays to identify drug targets in wastewater (just not festivals - but does that matter?).
- We deleted the sentence.
Line 333. It is worth mentioning that the technology used for this study (Nadarajan) is not widely available to other labs.
- We added this information.
Table 3. Considering the topic of the Table and heading of column (Preferred substances), the authors are advised to specify the parent drug, not the metabolite to avoid misinterpretation of the results.
- We changed the heading of the column to “preferred drugs”.
Line 548-549. Sampling frequency is only a consideration when sewer infrastructure is in place. In urinals, portable toilet blocks or portaloos, it is more about drug residue stability. Once excreted, the sample remains there until the unit is emptied – typically after the event or at night.
- We thank the reviewer for this summarizing explanation. We have inserted the following sentence accordingly: “Sampling frequency can only be considered in a meaningful way when sewer infra-structure is in place. In urinals, portable toilet blocks or portaloos, drug residue stability is the decisive factor. Once excreted, the sample remains in the collecting container until the unit is emptied – typically after the event or at night” (section 4.1.1).
Line 551. This is a sweeping statement that is carried forward throughout the manuscript. In fact, portable toilet blocks are often designated male/female and could be sampled when such infrastructure is in place. This obviously varies from country to country and festival to festival, but the authors should not generalise.
- We deleted the sentence: “The consistent adoption of time-proportional sampling approaches, typically collecting samples at 15-minute intervals (Brandeburova et al., 2020; Nadarajan et al., 2024), demonstrates recognition of the temporal variability inherent in festival consumption patterns, yet raises questions about whether current sampling frequencies adequately capture rapid fluctuations in substance use during peak consumption periods.”
Line 572. This is open to debate. The equipment used for direct injection is cost prohibitive for most labs and requires constant maintenance to maintain sensitivity. I challenge this approach as universally applicable or recommended. A SPE manifold cost next to nothing and can be operated by a person with minimal training. It is the mass spec that requires specialist operators and resources.
- We rephrased the whole paragraph to: “Nadarajan et al. (2024) suggested that direct injection methods can achieve comparable sensitivity and that this might have implications for the scalability and accessibility of festival monitoring programs, particularly in resource-limited settings where sophisticated-ed sample preparation infrastructure may not be available. However, the frugality of direct injection methods is debatable as the equipment used for direct injection is cost prohibitive for most laboratories and requires constant maintenance to maintain sensitivity. SPE manifolds, in contrast, are comparably cheap and can be operated by a person with minimal training. The mass spectrometer, however, requires specialist operators and re-sources” (section 4.1.1, last paragraph).
Line 685. The authors recognise the limitation of their choice of databases. I suggest adding SCOPUS.
- We did so.
Line 706. While this was clearly identified as a limitation in the literature search, it can be overcome with suitable infrastructure. The authors may include that as a future direction (line 814). Please note that movable toilet blocks are standard at festivals in parts of the world and not innovative at all.
- We inserted the sentence: “However, female underrepresentation and the lack of knowledge on gender differences can be overcome in the future with suitable infrastructure and technical adaptations” (section 4.2.3, last paragraph).
Line 734-738. This point is highly relevant. The authors should also mention the use of sanitiser chemicals in portaloos.
- We fully agree and inserted the sentence: “Additionally, the stability of target molecules might be impacted by contamination of the samples with sanitizer or deodorizer” (section 4.2.4, last sentence).
Line 741. The authors should perhaps consider if the lack of commercially available NPS is really such an issue. Whether a NPS is excreted or disposed of at the site, it shows the drug was intended to be used or could be used by other festival attendees. Isn’t the harm messaging the same?
- We toned our formulation down to “Another potentially critical aspect is the limited availability of commercial reference standards for NPSs”, and we deleted the words “consistently reported” to make this point less of an issue in section 4.2.5.
Line 771-779. Another point to consider is the rate of drug clearance from the body. During a festival event spanning less than a day, there is a high chance that drugs used during the festival may only be metabolised and excreted after the event, at a different site. In other words, interpreting results become complicated. It is also possible that drugs excreted at the site were consumed well prior to the event. Drug loads in wastewater from festivals should be interpreted with caution.
- We inserted the paragraph: “Another point to consider is the rate of drug clearance from the body. During a festival event spanning less than a day, there is a high chance that drugs used during the festival may only be metabolized and excreted after the event, at a different site. It is also possible that drugs excreted at the site were consumed well prior to the event. Drug loads in wastewater from festivals should therefore be interpreted with caution” (section 4.2.6.)
Line 834-838. I generally disagree with the statement about real-time monitoring, assuming the authors are alluding to wastewater analysis. Considering the analytical techniques described in this review required to measure the types of harmful drugs, real-time detection using wastewater is simply not possible. Additionally, it takes time for drugs to get metabolised and cleared from the body into the wastewater. The authors can calculate for themselves the time required for a typical LC/MS analysis and how few samples can be done in a day if sampling happens at a reasonable frequency.
- We deleted this statement without substitution.
Line 839. Please check the reference on the Glastonbury festival where the authors do discuss soil and water contamination.
- Even though the study did not meet our inclusion criteria, we dedicated a whole paragraph to Aberg et al. (2022), and we have additionally mentioned the study in section 4.1.3.
Reviewer 3 Report
Comments and Suggestions for Authors
The authors proposed a honest reviewing job. I regret that festivals in Asia (China...) were never studied but this review does not consider this.
I accept the publication of this article as it is.
Author Response
The authors proposed an honest reviewing job. I regret that festivals in Asia (China...) were never studied but this review does not consider this.
I accept the publication of this article as it is.
- We thank the reviewer for their positive assessment of our manuscript. In accordance with their comment, we have mentioned that the “Limitations include geographic underrepresentation of African and Asian countries … ” in the abstract of our amended manuscript.
Round 2
Reviewer 2 Report
Comments and Suggestions for Authors
I thank the authors for making changes and addressing all my comments.